# Structure and mechanism of the K$^+$/H$^+$ exchanger KefC

Ashutosh Gulati [1,2], Surabhi Kokane[1,2], Annemarie Perez-Boerema[1], Claudia Alleva [1], Pascal F. Meier[1], Rei Matsuoka[1] & David Drew [1] ✉

Intracellular potassium (K$^+$) homeostasis is fundamental to cell viability. In addition to channels, K$^+$ levels are maintained by various ion transporters. One major family is the proton-driven K$^+$ efflux transporters, which in gram-negative bacteria is important for detoxification and in plants is critical for efficient photosynthesis and growth. Despite their importance, the structure and molecular basis for K$^+$-selectivity is poorly understood. Here, we report ~3.1 Å resolution cryo-EM structures of the *Escherichia coli* glutathione (GSH)-gated K$^+$ efflux transporter KefC in complex with AMP, AMP/GSH and an ion-binding variant. KefC forms a homodimer similar to the inward-facing conformation of Na$^+$/H$^+$ antiporter NapA. By structural assignment of a coordinated K$^+$ ion, MD simulations, and SSM-based electrophysiology, we demonstrate how ion-binding in KefC is adapted for binding a dehydrated K$^+$ ion. KefC harbors C-terminal regulator of K$^+$ conductance (RCK) domains, as present in some bacterial K$^+$-ion channels. The domain-swapped helices in the RCK domains bind AMP and GSH and they inhibit transport by directly interacting with the ion-transporter module. Taken together, we propose that KefC is activated by detachment of the RCK domains and that ion selectivity exploits the biophysical properties likewise adapted by K$^+$-ion-channels.

Bacteria possess a number of transporters and channels to transport potassium (K$^+$) under a variety of different environmental conditions to maintain cell homeostasis[1–4]. One such system is the Kef transporters belonging to the monovalent cation:proton antiporter two (CPA2) superfamily[3–6], which exchange intracellular K$^+$ for external H$^+$ in response to electrophilic stress[7]. The Kef transporter domains are distantly related to the Na$^+$/H$^+$ antiporters[8,9], which help to maintain intracellular pH, sodium levels, and cell volume homeostasis[10,11]. In plants and algae, the Kef transporter equivalents (KEA1-6) appear to have a more similar role as the Na$^+$/H$^+$ antiporters as they can fine-tune the pH of chloroplasts or plastids[12–14]. In *Arabidopsis thaliana*, for example, the K$^+$ efflux antiporter (KEA3) is critical for high photosynthetic efficiency under fluctuating light[15,16]. Other KEA4-6 members are localized to endomembranes where they are important for pH regulation[17]. In animals, the Kef K$^+$/H$^+$ homologs belonging to the Transmembrane and coiled-coil domains 3 (TMCO3) family have been

linked to eye disease[18] and cancers[19], but their physiological role is yet to be determined[20].

KefB and KefC are the two main Kef transporters in *E. coli* and share 42% sequence identity (Supplementary Fig. 1)[7]. The KefB and KefC transporters from *E. coli* are also known as a glutathione (GSH)-gated K$^+$ efflux systems as their activity are regulated by GSH[5,21] (Fig. 1a). When the bacterium *E. coli* is overloaded by electrophiles, the formation of GSH adducts (GSX) activates either the KefB or KefC transporter, depending on the electrophile[5,22,23]. Although they have a similar role, the KefC system is currently better characterized[7]. Activation of KefC leads to short-term cytosolic acidification of 1–2 pH units, which in turn protects against electrophilic damage[23,24]. This K$^+$-efflux response pathway allows *E. coli* to resist and detoxify harmful metabolites. In order to activate KefC under the appropriate conditions, the KefC antiporter contains a C-terminal RCK domain[25], which binds AMP and either GSH or GSH-adducts[26]. Several structures of the

[1]Department of Biochemistry and Biophysics, Science for Life Laboratory, Stockholm University, SE–106 91 Stockholm, Sweden. [2]These authors contributed equally: Ashutosh Gulati, Surabhi Kokane. ✉e-mail: ddrew@dbb.su.se

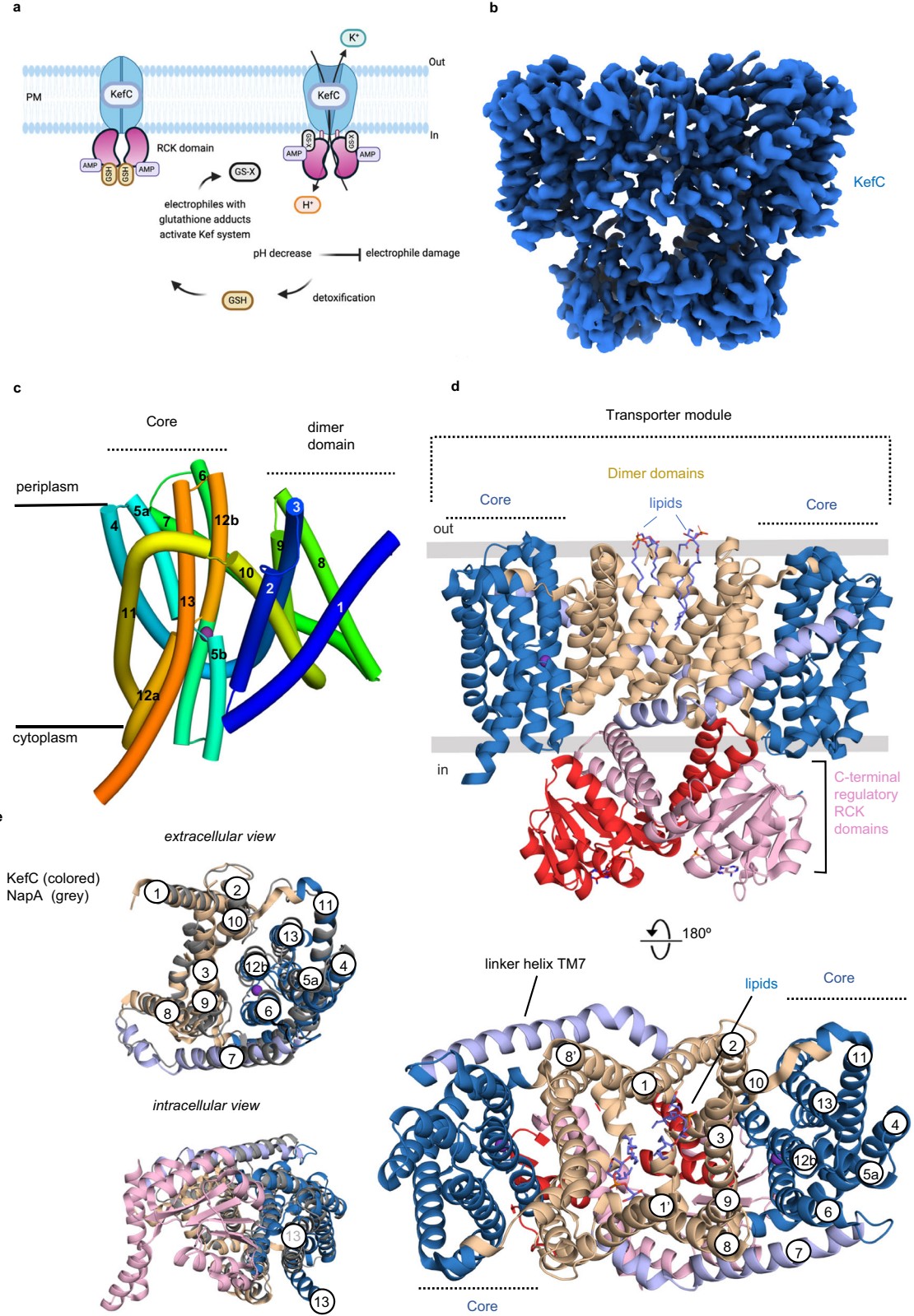

soluble KefC regulatory domains have been determined[25] and were observed as homodimers in complex with AMP, rather than the tetrameric assembly observed in RCK-containing K$^+$-ion channels[27]. The KefC transporter domain was expected to also form a homodimer, with a similar architecture to Na$^+$/H$^+$ exchangers[8], which consist of a scaffold domain and an ion-translocating core domain[28–30]. The core domain is typified by a six transmembrane (TM) bundle harboring two

opposite-facing discontinuous helices that crossover near the center of the membrane[30]. The closest structurally characterized CPA members to KefC is the NapA clade[8,9], for which crystal structures have been determined in both outward- and inward-facing conformations[28,31]. NapA and Na$^+$/H$^+$ exchangers in general operate by an elevator alternating-access mechanism[28,29,31–33]. In the elevator-model the scaffold domain mediating homodimerization, acts as an anchor for the

**Fig. 1 | The overall architecture of the K⁺/H⁺ exchanger KefC WT*. a** Schematic showing that the KefC protein is made of up a transporter module and a C-terminal nucleotide-binding regulatory RCK domain. The tripeptide glutathione (GSH) can accumulate up to concentrations exceeding 10 mM in *E. coli* to buffer oxidative stress and it inhibits KefC in the low mM range. The current working activation mechanism is that glutathione-dependent detoxification by formation of glutathione adducts activates KefC, results in the rapid efflux of potassium and a decrease in the intracellular pH. The decrease in intracellular pH protects *E. coli* against the toxic effects of electrophiles, e.g., methylglyoxal[85]. Figure 1a was created with BioRender.com released under a Creative Commons Attribution-NonCommercial-NoDerivs 4.0 International license https://creativecommons.org/licenses/by-nc-nd/4.0/deed.en **b** Cryo-EM density map of the KefC WT* homodimer colored in blue. **c** Cartoon representation of the KefC WT* monomer, which is made up of a 6-TM core domain and a scaffold domain, The core domain has two broken helices TM5a-b (teal) and TM12a-b (orange), which cross-over in the center of the membrane and form the ion-binding site location (purple sphere). **d** Cartoon representation of the KefC WT* homodimer shown from the membrane plane (above) and the extracellular side (below) with the ion translocation 6-TM core transport domains (blue), dimerization domains (sand) and linker helix TM7 (light purple). The C-terminal regulatory RCK domains (pink and red) interact between core and dimer domains and lipids bind between the dimerization interface (blue sticks). **e** Cartoon representation of the 13-TM KefC monomer (colored), superimposed onto the 13-TM inward-facing structure of NapA (gray) (PDB: 5BZ2).

highly mobile ion-transporting core domains[32]. Despite extensive characterization, however, an ion-bound state is yet to be observed. Here, we establish the mechanistic basis for K⁺ selectivity between Na⁺/H⁺ vs. K⁺/H⁺ exchangers, and provide a framework for how the activity of KefC is regulated by its nucleotide-binding RCK domains.

## Results and discussion
### Cryo-EM structures of KefC in complex with AMP at 3.2 Å resolution

Based on crystal structures of the KefC RCK domain and disorder prediction alogorithims[25,34], a KefC construct lacking the last 60 residues (referred to as WT*) was expressed and purified in the detergent LMNG, but was susceptible to aggregation (Supplementary Fig. 2a). We were able to purify stable KefC WT* by switching from NaCl to KCl salt, changing to the detergent DDM, and by the addition of the nucleotide AMP in all purification buffers (Supplementary Fig. 2b–d, see Methods). For structural studies of KefC WT*, DDM detergent in the buffer was replaced with the glyco-diosgenin (GDN) using size exclusion chromatography and grids were prepared at pH 7.5 (Methods). We collected around 25,000 movies and an EM map was reconstructed to ~3.1 Å with C1 symmetry according to the gold-standard Fourier shell correlation (FSC) 0.143 criterion, which contained ~305,700 particles (Table 1 and Supplementary Fig. 3a, b). The EM maps were well resolved for all the TM helices and loops except for a 14 amino acid stretch linking the transporter module with the soluble RCK domains (Fig. 1b, Supplementary Fig. 4, Methods). There was a further sub-class of KefC WT* that was reconstructed to ~3.9 Å with C2 symmetry according to the FSC 0.143 criterion, which contained ~141,700 particles (Supplementary Fig. 3a). However, in this 3D reconstruction there was discontinuous and weak map density for the dimerization interface and was therefore not used for detailed structural analysis.

The KefC monomer consists of 13 TMs with an extracellular N-terminus and intracellular C-terminus (Fig. 1c). The KefC WT* structure was more similar to bacterial Na⁺/H⁺ exchangers structures with 13 TM[28,29,33,35,36] (Fig. 1d, e), rather than NhaA from *E. coli* with 12 TMs[37,38] (Supplementary Fig. 5a). The KefC homodimer is formed predominantly by tight interactions between TM1 on one monomer and TM8 on the other, burying a total interface area of ~1600 Å² (Fig. 1d). Although sharing a closer evolutionary relationship with the Na⁺/H⁺ exchanger NapA[9], there is a larger hydrophobic gap between the KefC protomers, as that seen in the CPA1 members *Pa*NhaP and *Mj*NhaP[36,39] (Fig. 1d, e). As observed in several other Na⁺/H⁺ exchanger structures[33,39,40], density consistent with several lipids could be modeled at the dimerization interface (Fig. 1d). *E. coli* membranes are principally made up from 75% PE: 20% PG: 5% CL lipids but, lacking positively charged residues at the interface that could coordinate cardiolipin[41], we focused on the thermostabilization of detergent purified KefC WT*-GFP upon the addition of either synthetic PE, PG or PC lipids (Methods and Supplementary Fig. 5b). The clearest thermostabilization of the KefC homodimer was observed upon addition of PG lipids, and consequently two PG lipids were modeled into the cryo-EM maps, which lacked sufficient map density for their unambiguous

assignment (Fig. 1d and Supplementary Fig. 4). The cytosolic RCK domains span nearly the full length of the transporter domains. The core of the RCK domains is made up of the α/β Rossmann-fold[25], with two additional domain-swapped helices interacting with the TM ends of the transporter (Fig. 1d).

### The potassium site in the inward-facing conformation of KefC
Using FoldSeek[42], the KefC WT* monomer shows highest structural similarity to a NapA crystal structure that was trapped by disulphide residues in an inward-facing conformation (Fig. 1e)[43], with an overall rmsd of 3.6 Å for all Cα pairs. The regions with the poorest structural superimposition are the intracellular ends of TM12 and TM13 in the core domain. Moreover, the linker helix TM7 has an additional interfacial helical stretch, which has not been observed in NapA or other Na⁺/H⁺ exchanger structures (Fig. 1e). These structural differences collectively indicates that the KefC transporter module adjusts for the RCK domains, and indeed, recombinant exclusion of the C-terminal tail domain yielded unstable and aggregation prone constructs of KefC (Supplementary Fig. 5c). Consistent with an inward-facing structure, a highly negatively-charged funnel extends from the ion-binding site to the cytoplasmic surface (Fig. 2a), although the cavity entrance is partially blocked by the RCK domains. Near the base of the cavity, a strictly conserved aspartate residue D156 (TM6) is located, which corresponds to D157 in NapA, the primary acidic residue essential for ion-binding and transport in Na⁺/H⁺ exchangers[11,28,30,31,39,44].

The core domain contains two discontinuous helices TM5a-b and TM12a-b that cross over each near the center of the membrane (Fig. 2b). In KefC, the extended peptide breaks are either 3 or 4 residues each in length, creating a more close-knit helical cross-over than that seen in either bacterial or mammalian Na⁺/H⁺ exchangers (Fig. 2b and Supplementary Fig. 5d). Similar to NapA[28], a glutamate residue is located at the start of TM12b (E337) to neutralize the two positively-charged half-helical dipoles (Fig. 2b). Interestingly, the negatively-charged dipoles are neutralized by a lysine residue (K307) located on TM11, rather than an arginine residue in TM12a-b, as seen in bacterial and mammalian Na⁺/H⁺ exchangers (Fig. 2b and Supplementary Fig. 5d, e)[29,31]. Unexpectedly, the lysine residue K307 forms hydrogen-bonds to residues located on both TM5a-b and TM12a-b breakpoints (Fig. 2b). In NapA, the equivalent TM11 lysine (K305) doesn't form interactions with the half-helices, but instead makes a salt-bridge to an aspartic acid (D156) that is part of the so-called "DD" ion-binding motif[9,28]. In KefC, the "DD" motif is replaced by a "QD" motif[8] and the glutamine residue (Q155) that was expected to form an interaction with K307[8], is located some 6 Å away. The large distance between Q155 and K307 residues, as well as the lack of a charged interaction to K307, makes it unlikely that K307 would rearrange during the transport cycle, as modeled to occur upon Na⁺ binding and salt-bridge breakage in electrogenic Na⁺/H⁺ exchangers[38,43]. Thus, the ion-binding site is consistent with KefC and KEA plant homologs annotation as electroneutral ion-exchangers[45], i.e., since there is no charged interaction with the lysine to enable it to act as an additional H⁺ carrier[43].

**Table 1 | Data collection, processing, and refinement statistics of KefC structures**

| | KefC WT*(8BXG) | KefC WT*-GSH (8BY2) | KefC-GDN-D156N (9EMB) |
|---|---|---|---|
| **Data collection and processing** | | | |
| Magnification | 135,000 | 135,000 | 135,000 |
| Voltage (kV) | 300 | 300 | 300 |
| Electron exposure (e–/Å²) | 52.2 | 64.3 | 54.6 |
| Defocus range (µm) | 2.0–0.6 | 2.0–0.6 | 2.0–0.6 |
| Pixel size (Å) | 0.6645 | 0.6645 | 0.6645 |
| Symmetry imposed | C1 | C1 | C1 |
| Initial particle images (no.) | 7,550,000 | 10,657,172 | 15,028,339 |
| Final particle images (no.) | 305,737 | 260,115 | 326,951 |
| Map resolution (Å) | 3.16 | 3.18 | 2.98 |
| FSC threshold | 0.143 | 0.143 | 0.143 |
| Map resolution range (Å) | 2.7–4.5 | 2.8–4.5 | 2.6–4.5 |
| **Refinement** | | | |
| Initial model used (PDB code) | Alphafold model | KefC WT* structure | KefC WT* structure |
| Model resolution (Å) | 3. 1 | 3.1 | 3.0 |
| FSC threshold | 0.143 | 0.143 | 0.143 |
| Map sharpening B factor (Å²) | | –50 | –97 |
| **Model composition** | | | |
| Non-hydrogen atoms | 8507 | 8556 | 8505 |
| Protein residues | 1093 | 1094 | 1093 |
| Ligands | K: 2 | K: 2 | AMP:2 |
| | AMP: 2 | AMP: 2 | PGW:2 |
| | PGW: 2 | PGW: 2 | |
| | | GSH: 2 | |
| **B factors (Å²)** | | | |
| Protein | 64.48 | 81.56 | 115.52 |
| Ligand | 79.18 | 105.30 | 130.43 |
| **R.m.s. deviations** | | | |
| Bond lengths (Å) | 0.003 | 0.002 | 0.002 |
| Bond angles (°) | 0.535 | 0.521 | 0.581 |
| **Validation** | | | |
| MolProbity score | 1.67 | 1.60 | 1.87 |
| Clashscore | 4.29 | 4.78 | 4.69 |
| Poor rotamers (%) | 0.23 | 0.00 | 2.76 |
| **Ramachandran plot** | | | |
| Favored (%) | 92.63 | 94.94 | 95.76 |
| Allowed (%) | 7.37 | 5.06 | 4.15 |
| Disallowed (%) | 0.00 | 0.00 | 0.09 |

WT* reperesents a KefC construct lacking the last 60 residues.

We observed additional non-protein density next to the side-chains of Q155 and D156 residues, which was consistent with that of a bound K+ ion (Fig. 2c and Supplementary Fig. 6a). The K+ ion is coordinated within interaction distance of 2.6 to 3.1 Å by side-chains of Q155, D156, and T127. The K+ ion is also coordinated to the backbone carbonyl oxygen of S125, which is located between TM5a and TM5b helices. The K+ ion is further coordinated from above and below by longer-range polar interactions of 3.5–4.2 Å from the side-chain of S336 located between TM12a and TM12b (Fig. 2b and Supplementary Fig. 6b) and the carbonyl oxygen of L152 (Fig. 2c and Supplementary Fig. 6b). The side-chain of S125 further interacts directly with K307,

which stabilizes the ion-binding site (Fig. 2d). We attempted to purify a K307A variant, to assess its role, however the protein was too unstable and only monomers could be extracted in detergent (Supplementary Fig. 6c). The ion-binding site is enclosed above and below by the hydrophobic side chains of L152 and F338 residues (Supplementary Fig. 6b). Hydrophobic residues in this position are conserved across all Na+/H+ antiporters[8], and their positioning next to a bound K+ ion is consistent with their proposed function as hydrophobic gates[29,46], controlling accessibility to the ion-binding site. To strengthen the assignment of the K+ ion, we substituted the strictly conserved ion-binding aspartate D156 in KefC WT* to asparagine, which in Na+/H+ antiporters has been shown to be essential for ion-binding and transport[28,29,44]. Following a similar cryo-EM workflow, the cryo-EM map for the KefC D156N variant was resolved to ~3.0 Å resolution in the presence of 300 mM KCl (Supplementary Fig. 7a). The cryo-EM map is of similar quality to the previous maps obtained with 150 mM KCl, yet no density for any K+ ion was observed in the D156N variant (Supplementary Fig. 7b).

Despite a high-resolution structure of NapA at 2.2 Å resolution[31] and several other X-ray and cryo-EM structures of Na+/H+ exchangers[29,33], a Na+-bound state has yet to be experimentally observed. The Tl+ bound structure of PaNhaP showed that Na+ is likely to bind within the funnel entrance by interaction with the conserved ion-binding aspartate[39]. However, the Tl+ ion in PaNhaP was also found to form additional interactions to the dimer domain via a negatively-charged residue, which is not conserved or required for transport[39]. Moreover, a repositioned Na+ site situated between the core and dimer domains would be inconsistent with an elevator mechanism, which is defined by the fact that the substrate is translocated only by one of the domains[32]. In KefC, however, the K+ ion is enclosed entirely within the core domain, sandwiched between the strictly conserved "QD" motif (Fig. 2c and Supplementary Fig. 8) and interacting with residues located on both unwound regions (Fig. 2c and Supplementary Fig. 6b). The location of the K+ ion coordination is consistent with MD simulations of Na+ binding to inward open NapA[31]. The ion-binding site seems to be pre-formed in KefC as there is no obvious difference in the ion-binding site residue positions between the D156N and WT* structures (Supplementary Fig. 7b). This contrasts to the Na+ coordination seen in MD simulations of NapA[31], PaNhaP[46], NhaA[38], NHE9[29] and NHA2[40] Na+/H+ exchangers, where Na+ coordination required rearrangement of ion-coordinating residuesand an additional 2 to 3 water molecules.

It seems that the more rigid ion-binding site in KefC is well-suited for the coordination of a dehydrated K+ ion, and would disfavor binding of a Na+ ion due to the higher energetic penalty associated with removal of water from the smaller cation[47]. To strengthen this hypothesis, we performed alchemical perturbation simulations for the bound ion in the KefC WT* dimer embedded in a model membrane bilayer (see Methods). We observed a binding free energy difference ($\Delta\Delta G_{Na\to K}$) for Na+ versus K+ with a calculated selectivity towards K+ of ~14 KJ/mol, compatible with previous results estimated in K+ selective sites[48,49] (Supplementary Fig. 9a–c). This K+ selectivity rationale is consistent with the replacement of the "ND/DD" found in Na+/H+ exchangers with the "QD" motif[9], since the longer side-chain amide of Q155 forms an additional hydrogen bond to S125 in the TM5a-b break, which would better accommodate the partly dehydrated K+ ion (Fig. 2d).

## SSM-based electrophysiology of KefC

To further probe ion selectivity of KefC WT* protein was reconstituted into liposomes made from E. coli polar lipids for solid-supported membrane (SSM)-based electrophysiology recordings, which is a more sensitive technique than patch-clamped electrophysiology for low turnover transporters[50–52]. In this technique, proteoliposomes are adsorbed to a SSM, and charge translocation of ions is measured via

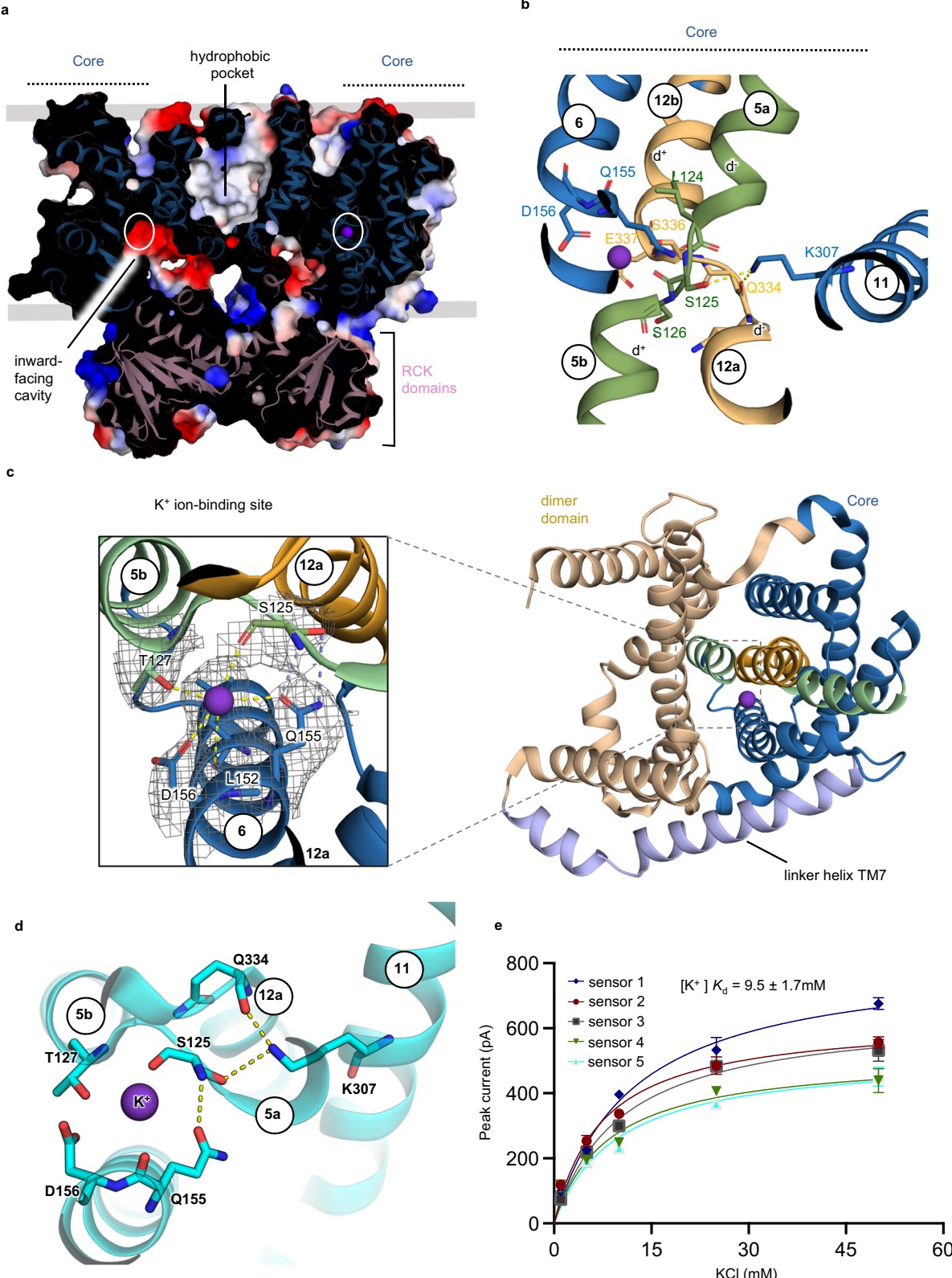

**c** K⁺ ion-binding site

capacitive coupling of the supporting membrane[50]. SSM-based electrophysiology can also detect pre-steady state currents of the half-reaction in electroneutral transporters. Peak currents at pH 7.8 were observed for KefC upon increasing concentrations of K⁺ that were fairly low at less than 1 nA, but clearly higher than the signal obtained from the ion-binding aspartate D156N variant or empty liposomes (Supplementary Fig. 9d).

The low peak currents indicated that the K⁺-induced response could arise predominantly from ion binding events, rather than transport. Consistently, K⁺-induced currents recorded upon various lipid-protein-ratio (LPRs) showed that the associated decay currents had similar decay time constants, which would not be the case if a charge build up had occurred with transported K⁺ ions (Supplementary Fig. 9e). The apparent binding affinity of KefC WT* for K⁺ was

**Fig. 2 | Structure of the K⁺ ion-binding site and its electroneutral activity.**
**a** Cartoon representation of dimeric KefC from the membrane plane with the electrostatic surface representation rotated to more clearly show one of the inward-facing funnels (colored blue to red, for positive to negative charges, respectively). The strictly conserved ion-binding residue location for D156 is shown as a circle. **b** Cartoon representation of the KefC 6-TM core transport domain. The crossover of broken helices TM5a and TM5b (green) and TM12a and TM12b (light orange) have peptide breaks that are only 3 residues long and are much shorter than that seen in Na⁺/H⁺ exchanger structures. The half-helical dipoles are neutralized by oppositely-charged residues (E337 and K307), with a distinct direct interaction with the TM11 K307 interacting directly with one of the ion-binding residues (S125). **c** Cartoon of the KefC monomer (left) ion-binding residues (right) that are made up of the "QD" motif characteristic to the K⁺/H⁺ exchangers[7] and the corresponding cryo-EM maps showing the additional non-protein density for a

bound K⁺ ion (purple sphere). Dashed lines represent interaction distances at ~2.6–3.5 Å. The side-chain residues are arranged closer to each other than in Na⁺/H⁺ antiporters to coordinate a dehydrated K⁺ ion (yellow-dashed lines) as facilitated by shorter peptide breaks and Q155 forming hydrogen bond interactions (blue dashed-lines) to S125. **d** Top-down view of the potassium binding site of KefC from the extracellular side. Important ion binding site residues are shown in cyan sticks. Network of interactions stabilizing the short helical cross over are represented as yellow dashed lines. K307 present in the TM11 makes direct contact with residues present in TM5 and TM12 helical crossovers. **e** Fit of the transient peak currents as a function of K⁺ concentrations for KefC and the corresponding binding affinity ($K_d$). Error bars are the mean values ± s.e.m of $n = 3$ or 4 technical repeats. The $K_d$ was calculated from the average fitting from $n = 5$ individual titrations (sensors). The source data are provided as a source data file.

determined at $K_d$ = 8 to 10 mM using either choline chloride or NaCl in the non-activating buffers (Supplementary Fig. 9f and Fig. 2e). This is consistent with the reported apparent $K_M$ of ~4 mM in vesicles directly isolated from an *E. coli* Na⁺(K⁺)/H⁺ antiporter transporter deletion strain[4] and its proposed physiology of operating in the low-to-high mM K⁺ concentrations[7].

The K⁺ binding affinity measured by SSM for KefC WT* is similar in the presence and absence of Na⁺, which indicates that KefC is K⁺ selective. Na⁺/H⁺ exchangers typically bind Li⁺ with higher affinity than Na[+40,43]. To confirm K⁺ selectivity, NaCl or LiCl were added to KefC WT* liposomes using 200 mM choline chloride in the non-activating buffers. Although the peak currents upon Na⁺ and Li⁺ addition were similar to K⁺, there was no clear saturation. Also, the signals observed were similar to that measured for the D156N variant, indicating peak currents are a result of non-specific binding, i.e., high salt concentrations may trigger protein rearrangements that gives rise to charge displacement (Supplementary Fig. 9g). An ion-binding glutamine-to-aspartic acid variant to match electrogenic Na⁺/H⁺ antiporters[43], has proven to be non-functional for both K⁺ and Na⁺ in plant KEA homologs[45]. We repeated SSM measurements for the Q155D KefC variant and likewise found that it abolished K⁺ binding (Supplementary Fig. 9f). The residue corresponding to T127 in KefC is a valine in NapA, which in KefC provides a hydroxyl group for K⁺ coordination. We substituted T127 to valine, but the variant showed no clear binding for either Na⁺ or K⁺ (Supplementary Fig. 9f, g). Taken together, SSM-based electrophysiology data are consistent with the selectivity of KefC for K⁺ and, as somewhat expected, single variant substitutions seem insufficient to alter ion specificity.

**The C-terminal regulatory RCK domains of KefC**
Using FoldSeek[42] to search for the most similar RCK domains[53] to those in KefC, showed that the closest experimental structures were RCK domains found in the bacterial K⁺ ion-channels MthK, KtrAB, and TrkH[7,54–56] (Supplementary Fig. 10a, b). RCK domains are also referred to as KTN (K⁺ transport, nucleotide-binding) domains based on their homology to dinucleotide-binding motifs[56]. The regulatory domain of TrkH for example, referred to as TrkA, forms tetramers in the presence of ATP[55]. The RCK domains of KefC have been extensively analyzed in isolation and have been shown to co-purify with AMP[7,25,26]. The crystal structure of the soluble RCK domains with AMP bound are dimeric[26], matching the KefC homodimer (Fig. 3a and Supplementary Fig. 11a). Like TrkA, each RCK domain has a conserved Rossmann-fold where the nucleotide binds, followed by a short helix that joins the two subunits together (Fig. 3a and Supplementary Fig. 4). The coordination of AMP is overall consistent with the crystal structure of the soluble RCK domains of KefC bound to AMP from *E. coli*[34] and the bacterial homolog *Shewanella denitrificans*[26] (Supplementary Fig. 11a). The nucleobase adenine forms π-cation interactions to H430, the primary amine is coordinated by D449, and the imine group to D472, which further coordinates the ribose sugar together with D429 (Fig. 3a). The

terminal mono-phosphate is coordinated by two arginine residues R409 and R496 that, all together, creates a tight binding pocket for AMP, consistent with the reported thermostabilization of KefC soluble domains at low concentrations[26].

In contrast to TrkA and KtrB, however, the RCK domains do not bind ATP[26] and the RCK domains in KefC have an additional terminal helix (α7) (Fig. 3b and Supplementary Fig. 10b). The domain-swapped α6 and α7 helices are mediating most of interactions with the transporter module in KefC (Fig. 3a, b). A superimposition of the KefC monomers shows that the regulatory RCK domains adopt different conformations (Supplementary Fig. 11b). Consequently, the RCK domains form altered transporter-RCK interactions (Fig. 3b). The asymmetric RCK-transporter interactions imply that there is some flexibility in how the RCK domains interact with the transporter module. Nevertheless, the same interactions are observed in both WT* and the D156N KefC structures (Supplementary Fig. 11c) and the local resolution estimates of the RCK domain are overall similar to the transporter domain (Supplementary Fig. 3a), indicating that the final conformations are stable. In one protomer, α6 and α7 helices lie parallel to the intracellular-facing surface of the transporter module, with several interactions formed between α7 and the transporter module, burying a surface area of ~480 Å² (Fig. 3b). In the other protomer, the RCK domain are rotated some 60°, and residues in α6 and α7 helices predominantly interact with the core domain, burying a contact area of ~780 Å² (Fig. 3b). In this protomer, charged residues E539 and R543 in α7 hydrogen-bond to N135 and N138 in TM5b. The residues R401 and E465, located at the start of β-sheet β1 and β4, hydrogen bond to R146 and T142 of TM6 in the core domain, respectively. Further interactions between S532 and E531 of α6 helix with K81 and H259 residues are also observed. The hydrogen bond interactions between R543 and N135 are the only conserved interactions in both protomers, with an additional interaction observed between N553 in α7 and the backbone of R212 in TM7-TM8 loop. The distinct amphipathic helix connected to TM7 harboring four arginine residues – R200, R204, R208, and R212 – could further help to stabilize interactions with the RCK domains (Fig. 3a, b).

In the current KefC structures, multiple interactions are formed between the RCK domains and the transporter module, and thus likely represent an inactive state of the transporter. Nevertheless, under more physiological conditions, GSH should also be bound, as it is present at mM (~10 mM) concentrations in the cytoplasm[57]. To establish if GSH would alter how the RCK domains interact with KefC, we further determined the cryo-EM structure of KefC in the presence of 20 mM GSH to 3.2 Å resolution (Fig. 3c, d, Table 1, and Supplementary Fig. 12a). Comparing the AMP only with the AMP/GSH cryo-EM maps, we observe additional density for GSH in the highly positively-charged cleft proximal to the bound AMP molecules (Fig. 3c, d and Supplementary Fig. 12b). The GSH coordination is consistent to that seen in the crystal structure of the soluble RCK domains of KefC[34] (Supplementary Fig. 13a). In particular, the GSH interacts between R409 and Q412 in one protomer and R498 and R516 of the other (Fig. 3c).

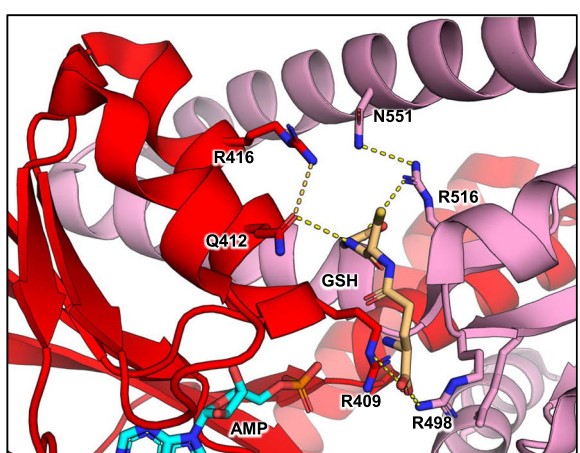

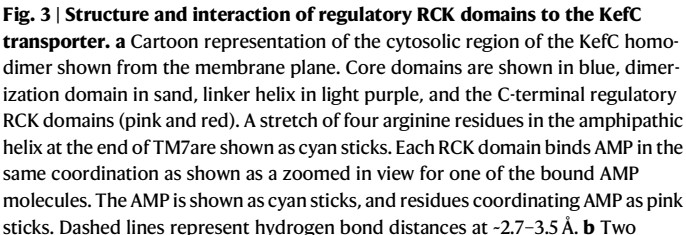

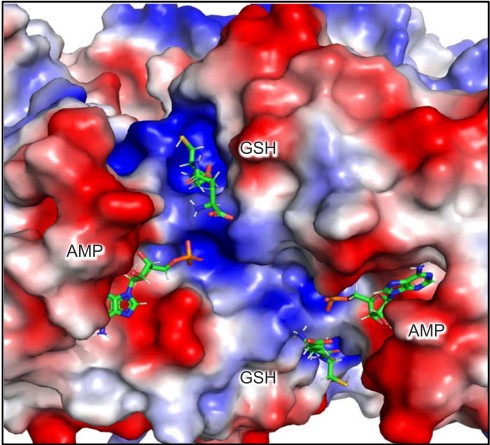

**Fig. 3 | Structure and interaction of regulatory RCK domains to the KefC transporter. a** Cartoon representation of the cytosolic region of the KefC homo-dimer shown from the membrane plane. Core domains are shown in blue, dimerization domain in sand, linker helix in light purple, and the C-terminal regulatory RCK domains (pink and red). A stretch of four arginine residues in the amphipathic helix at the end of TM7 are shown as cyan sticks. Each RCK domain binds AMP in the same coordination as shown as a zoomed in view for one of the bound AMP molecules. The AMP is shown as cyan sticks, and residues coordinating AMP as pink sticks. Dashed lines represent hydrogen bond distances at ~2.7–3.5 Å. **b** Two interaction sites between parallel domain-swapped helices in the RCK domain and the KefC transporter module are shown. Side-chains are labeled and shown as sticks with dashed lines representing hydrogen bond interactions. **c** Interactions involved in binding of GSH (shown in sand sticks). GSH molecules binds at the interface of dimeric RCK domain. Interacting residues from individual protomers are shown in pink and red sticks, respectively. **d** Electrostatic surface representation of the RCK dimer as viewed from the cytoplasm with bound AMP (cyan) and GSH (sand) shown in stick.

Moreover, R416 hydrogen bonds to Q412 and may indirectly influence GSH coordination, which would be consistent with the observation that R416S and Q412K variants abolished GSH inhibition[34]. It is plausible that the GSH positioned 5.7 Å from the terminal phosphate in AMP would preclude the longer ATP from binding, as seen in RCK-regulated bacterial K+-channels (Fig. 3c). The residue R516 further hydrogen bonds to N551 located in α7, which is the helix characteristic to KefC that forms many of the interactions to the transporter domain (Fig. 3c). Consistent with a coupling between the two domains, 3D variability analysis (3DVA) reveals multiple positions for the RCK domains (Supplementary Movie 1). States where α7 in the RCK domain interacts most extensively with the linker TM7 in the transporter domain, are associated with stronger map density for GSH (Supplementary Fig. 13b). The 3DVA of the AMP only structure shows less RCK domain mobility, although the final conformation was similar (Supplementary Movie 2). Moreover, in *E. coli* cells the R416S variant abolishing GSH inhibition is thought to create a constitutively active transporter[20]. It seems that the negatively-charged GSH peptide may further stabilize the energetically unfavorable interfaces of positively-charged residues, which come together upon AMP-induced dimerization of the RCK domains (Fig. 3d).

Since the soluble RCK domains natively co-purify with AMP, under resting conditions, it is likely that AMP inhibits ion-exchange in KefC by stabilizing RCK dimer formation, which then interacts with the transporter module. Consistent with RCK domains interacting with the transporter domains when AMP is present, heating detergent-solubilized KefC-GFP membranes at 40 °C for 10 min breaks the protein into monomers in the absence of AMP, with no stabilization seen for the addition of either CMP or cyclic di-AMP nucleotides (Supplementary Fig. 13c). Remarkably, incubating KefC with shrimp alkaline phosphatase to remove the monophosphate in AMP at 4 °C for 30 min, is enough to fully shift the KefC protein to a monomer in detergent, confirming the RCK domains are likely to only interact extensively with the transporter module when AMP is bound (Fig. 4a). To examine the RCK-transporter module interactions in more detail, further KefC point mutations were constructed and their stability in detergent-solubilized membranes were compared to KefC WT* in the presence of AMP. Single point mutations of either H259A, E465A, or R543A completely disrupted homodimer retention after heating, indicating the variants weakened RCK interactions with the transporter module (Fig. 4b, c). On the other hand, N135A R146A, and R401 variants retained a higher homodimer fraction after heating than KefC WT*, indicating that the RCK domain interactions were stronger (Fig. 4b, c). The addition of shrimp alkaline phosphatase to the R146A variant nonetheless shifted the protein to a monomer, confirming that the interactions were still AMP dependent (Supplementary Fig. 14a). It is possible that neutralization of these charged residues shifts the equilibrium to a more homogenous population, although this is speculative. Nevertheless, we can confirm that KefC stability is directly coupled to the presence of AMP/RCK dimer formation. Consistent with this analysis, we further collected cryo-EM data for KefC WT* protein that was dialyzed to remove AMP at the last purification step. Based on the 2D classes we no longer observe any features for the RCK domains and, likely due to transporter flexibility, we were no longer able to build a homogenous 3D reconstruction for KefC WT* (Supplementary Fig. 14b).

## Conclusions

Potassium is an important cation across all Kingdoms, and life has evolved a sophisticated network of various K+ transport systems for over millions of years. Bacteria possess a number of K+ channels for enabling the cation to go down its electrochemical gradient upon selective stimuli (e.g., KtrAB) and both primary- and secondary-active transporters for its influx e.g., KdpFABC and KUP system[1]. The KefC system is the major energized transporter for K+ efflux and together

with the KEA1-6 homologs in plants[13,16], they have a critical role in pH regulation, detoxification, and osmoregulation. Indeed, with up to 10% of its dry weight, K+ is the most abundant cation found in plants, and deletion of *kea1kea2* genes leads to a dramatic decrease in the proton-motive-force in thylakoid membranes[58]. The evolutionary relationship between the K+/H+ exchangers and the Na+/H+ exchangers[8,9] implies that they will share a similar mechanism.

Here, the structure of KefC confirms the overall expected NhaA-fold[37], but with some unforeseen differences in the core domain with cross-over helices that are much closer together. The bound K+ ion was unexpected as we are yet to observe a bound Na+ ion in any of the previously determined Na+/H+ exchanger structures[28,29,31,33,36,38–40,59]. We have previously proposed that Na+ binding in the Na+/H+ exchangers will neutralize the strictly conserved ion-binding aspartate, so that it can by-pass the hydrophobic surface of the dimer domain[31]. Given the fast turnover of Na+/H+ exchangers at 1500 s⁻¹ [30], the substrate-induced conformations are likely to be minimal, and the ion binding site only transiently occupied. It is plausible that we have been able to observe a K+ ion in KefC as its coordination has no mediating waters, and the core domains are largely restricted from moving by the RCK domains. Importantly, the ion-bound state confirms that the core domain in cation:proton antiporters (CPA) are able to completely enclose their substrate ion, consistent with their annotation as elevator transporters[32]. The bound K+ ion also demonstrates how the ion is coordinated in a dehydrated state, which is less favorable for Na+ due to the larger energetic penalty for dehydration. The closer arrangement of ion-binding residues is facilitated by additional hydrogen bond between a serine residue and glutamine residue from the "QD" motif. Notably, ion-binding glutamine-to-aspartic acid variants that have been constructed to match the "DD" motif have proven to be non-functional. Taken together, we propose that the selectivity for K+ vs Na+ is established by both the architecture of the cross-over helices and the closer arrangement of side-chain residues.

Allosteric regulation is an important property of ion-channels and transporters. The C-terminal tail of mammalian Na+/H+ exchangers (NHEs) is a well-established regulatory element[11]. Unlike KefC, the C-terminal tail of NHE proteins have no obvious tertiary structure and, instead, contain secondary-structure elements and disordered regions, which can bind many different external proteins, e.g., Ca²⁺calmodulin (CaM)[60]. The structure of KefC is overall consistent with an auto-inhibitory model, whereby the RCK domains interact with the core domains to restrict their movement. This is in contrast to RCK containing K+ ion channels where a conformational rearrangement of the RCK domains triggers a conformational change in the channel[61]. Under standard conditions, in the presence of AMP and high intracellular levels of GSH, the RCK domains are close enough to interact with the KefC homodimer, forming multiple electrostatic interactions to inhibit transport activity.

KefC activity is greatly increased in response to glutathione-adducts, which are produced by *E. coli* in response to toxic electrophiles[4]. The electrophiles either react spontaneously with glutathione (GSH), or GSH is conjugated to the nucleophile by glutathione S-transferases, which is the first step during the electrophile detoxification. Glutathione adducts might in KefC displace GSH[21], which is a direct inhibitor. The activated structural state of KefC is unclear. A plausible mechanism is that glutathione adducts that bind more tightly ($K_d$ = 0.4 to 12 μM GSX vs 900 μM for GSH)[62] displace AMP, and cause the breakdown of RCK domains into monomers and their physical detachment from the transporter. Such a regulatory mechanism would be analogous to how TrkA domains in K+ channels undergo a tetramer-to-dimer conversion by switching between ATP to ADP binding, where the ADP bound state detaches from TrkH to enable K+ conduction[54]. However, mutations disrupting AMP binding have also been shown to abolish KefC activity[26], which would rather imply that the active state still retains AMP and, as such, RCK dimerization. Nevertheless,

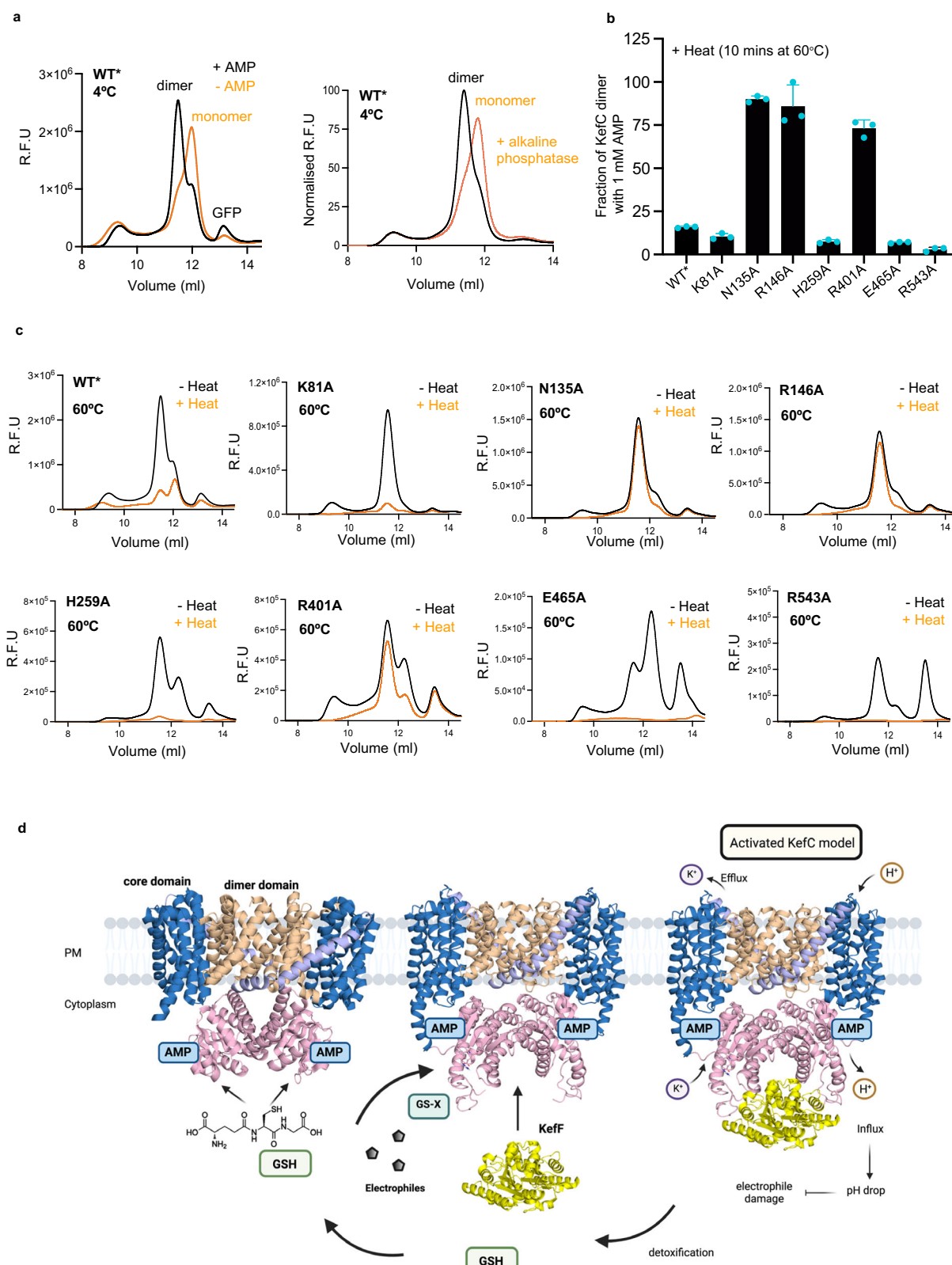

glutathione adduct binding to KefC must somehow facilitate the detachment of the RCK domains. In fact, in *E. coli* cells an accessory soluble NADH-binding protein KefF is required for maximal KefC activity, and in a *kefC kefF* deletion strain, only ~10% of the activity is observed upon complementation with KefC versus both KefF and KefC[4,22]. It is possible that the KefF protein is the additional factor required to stabilize detached RCK domains, yet the KefF protein is

reported to interact weakly with KefC[34] and we have likewise been unable to form complex with purified components (Supplementary Fig. 15a). By genetically fusing the KefF protein with the soluble RCK domains of KefC, it was possible to obtain a complex crystal structure in the presence of GSX[34]. In the complex crystal structure, the KefF protein interacted on the same surface interface as the interactions seen here between the RCK domains and the KefC transporter

**Fig. 4 | Effect of AMP removal on KefC. a** *Left:* representative FSEC traces for solubilized membranes of KefC WT* in the presence (black) and absence of AMP (orange) at 4 °C. *Right:* normalized FSEC traces of solubilized membranes of KefC WT* with AMP (black) and KefC WT* incubated with alkaline phosphatase (orange). Removal of AMP in both cases destabilizes KefC dimers. **b** Fraction of dimeric protein retained in DDM solubilized membranes of KefC and RCK-transporter module interface mutants after heating at 60 °C in the presence of AMP. Error bars indicate mean values ± sd of $n = 3$ independent experiments. The source data are provided as a source data file. **c** Representative FSEC traces for KefC WT*, K81A, N135A, R146A, H259A, R401A, E465A, and R543A at 4 °C (black) and after heating at 60 °C for 10 min (orange) in buffer containing AMP. **d** Schematic of the KefC

activation mechanism. Under resting conditions, AMP and GSH bind to the homodimer assembly of the RCK domains and the unique C-terminal helices (α7) interact with the transporter module inhibiting transport, as shown in the cryo-EM structure here. Upon displacement of GSH by glutathione adducts the RCK domains likely detach from the transporter module and interacts with KefF, which is predicted here based on the AlphaFold[71] KefC model. Detachment of the RCK domains allows the KefC transporter to operate according to ion-concentration gradients and the pKa of the ion-binding site, which is specific for K⁺ co-ordination ($K_d$ ~ 10 mM). Figure 4d was created with BioRender.com released under a Creative Commons Attribution-NonCommercial-NoDerivs 4.0 International license https://creativecommons.org/licenses/by-nc-nd/4.0/deed.en.

module[25] (Supplementary Fig. 15b). This complex crystal structure would be consistent with the hypothesis that the KefF protein is required to stabilize RCK domain detachment (Fig. 4d). Although, we cannot rule out the possibility that the KefC-KefF crystal structure complex has formed a non-physiological arrangement, we find it interesting that AlphaFold2[63] predicts a flipped conformation of the RCK domains in KefC (Supplementary Fig. 15c). The AlphaFold predicted aligned error between the transporter and the RCK domains is high, which suggests an ambivalent relative position of these two domains with respect to one another (Supplementary Fig. 15d). Additionally, the predicted local distance difference test (pLDDT) score for the linker residues is relatively low, which could be indicative of inherent flexibility (Supplementary Fig. 15c). We applied ColabFold[64] to predict the multimeric assembly of KefC and KefF, both with and without the KefC structure as a template. Surprisingly, both the predicted models present the flipped conformation of the RCK domain with bound KefF (Supplementary Fig. 15e). Taken together, although unconventional, it seems the most plausible activation model of KefC is the stabilized detachment of flipped RCK domains by the KefF protein, likely facilitated by GSX binding to KefC (Fig. 4d).

In summary, although further structures are required to determine the activated state of KefC, our work nonetheless provides a model framework for both ion selectivity and extrinsic regulation in K⁺/H⁺ exchangers, outlining concepts likely relevant to Na⁺/H⁺ exchangers and many other types of ion-transporters in general.

## Methods
### KefC cloning, expression and purification
The *kefC* gene was amplified by PCR using genomic DNA present in the cell lysate of *E. coli* MACH1 cells (Thermo Scientific). Nde1 and BamH1 restrictions sites were introduced during amplification to help with cloning. The KefC structural construct (WT*) is missing the final 60 C-terminal residues, which were not included due to predicted protein disorder and previous structural work on the RCK domain[25]. Sticky end overhangs were produced following digestion using NdeI and BamHI restriction enzymes (NEB) and ligated into the double digested pWaldo-GFPd vector[65] using T4 DNA ligase (NEB). The ORF also encodes for the expression of, downstream of insert: Tobacco Etch Virus (TEV) digestion site, followed by a C terminal GFP-His₈ fusion-tag. RCK deletion constructs (KefC-△379 and KefC-△399) were similarly made by amplifying respective gene strings from KefC structural construct and ligated into the pWaldo-GFPd vector. All single site variants were derived from the KefC structural construct and were made using the QuikChange lightning site-directed mutagenesis kit (#210519 Agilent) (Supplementary Table 1).

The KefC constructs were over-expressed in *E. coli* Lemo (DE3) strain (NEB) using the previously published MemStar protocol[66]. In brief, 10 L of 1 × MemStar media supplemented with 50 µg/ml kanamycin was inoculated with a saturated culture to a final OD₆₀₀ of 0.1 AU. The cultures were grown at 37 °C at 180 rpm until OD₆₀₀ of 0.6 AU. Protein overexpression was induced by addition of IPTG to a final concentration of 0.4 mM, and cell cultures were further incubated for 16 h at 16 °C. *E. coli* cells were harvested by centrifugation at 4000 x

g for 10 mins, supernatant discarded, and the cell pellet resuspended in buffer A containing 25 mM Tris pH 7.5, 300 mM KCl, and 1 mM AMP. Resuspended cells were lysed by mechanical disruption at 30 kPsi by two passes through a Constant Systems cell disruptor. Remaining cells and cell debris were removed by centrifugation at 10,000 × g for 15 min at 4 °C. The clarified supernatant was recovered and membranes were isolated by ultracentrifugation at 195,000 × g for 90 min at 4 °C. Membranes were resuspended in 50 ml of buffer A using a glass homogenizer.

For KefC purification, membranes were solubilized in 300 ml of buffer A supplemented with 1% (w/v) dodecyl-β-D-maltopyranoside (DDM, Glycon technologies) and stirred in a 500 ml beaker at 4 °C for 90 min. Non-solubilized membrane material was pelleted by ultracentrifugation at 195,000 × g for 45 min at 4 °C, and the supernatant containing the solubilized membranes were incubated with 10 ml of Ni-NTA agarose resin (Qiagen), pre-equilibrated with buffer A for 90 min. The Ni-NTA slurry was transferred to a 2.5 × 20 cm Econo-glass column (Bio-Rad) and washed with 3 × 200 ml of buffer containing 20 mM Tris pH7.5, 300 mM KCl, 1 mM AMP, 0.1% (w/v) DDM and either 30-, 40-, or 50-mM Imidazole, respectively. KefC was eluted with 40 ml elution buffer containing 20 mM Tris pH 7.5, 300 mM KCl, 1 mM AMP, 0.1% (w/v) DDM and 250 mM Imidazole. The eluted KefC-GFP fusion was digested with equimolar His₆-TEV protease during dialysis overnight at 4 °C against 3 L of 20 mM Tris pH 7.5, 150 mM KCl, 1 mM AMP, 0.03% DDM (w/v) in 3.5 kDA MWCO dialysis-tubing (Snake skin, Thermo scientific). The uncleaved GFP-His₈-tag and His₆-TEV proteins were immobilized using a 5 ml HisTrap column (GE healthcare), and the flow-through containing the digested KefC was collected and concentrated using a 100-kDa MW cutoff concentrator (Millipore) at 3000 × g, 4 °C. The purified KefC protein was injected onto an Enrich SEC 650 10 × 300 Column (Bio-Rad) pre-equilibrated with a buffer containing 20 mM Tris pH 7.5, 150 mM KCl, 1 mM AMP, 0.03% (w/v) DDM and the main peak collected. For Cryo-EM grid preparation, the protein was subjected to SEC in the same buffer replacing the DDM with 0.01% (w/v) glyco-diosgenin (GDN) detergent (Anatrace). All the other variants were also purified using the above-mentioned protocol.

### Cryo-EM sample preparation and data collection
Purified KefC was concentrated to 7 mg/ml and 3 µl of the final sample was applied to freshly glow-discharged QUANTIFOIL Cu300 R2/1 holey carbon grids. Samples were blotted for 3 s under 100% humidity at 4 °C and plunged frozen in liquid ethane using the Vitrobot Mach IV. Grids for D156N variant were also made using the same protocol. For samples used for data collection of the glutathione bound KefC structure, the KefC protein was incubated with 20 mM (from pH adjusted 250 mM stock) of glutathione for 1 h prior to grids preparation. Cryo-EM datasets were collected using Titan Krios G3i (Thermo scientific) microscope operated at 300 KeV and equipped with Gatan K3 Bio-Quantum detector. The datasets were collected at a magnification of 130,000x, with a pixel size of 0.6645 Å in a counted super-resolution mode. The movies were recorded with a defocus range of 0.4 to 2.0 µM using EPU2. Other data collection parameters and statistics are summarized in the Table 1.

## Cryo-EM data processing

All datasets were processed using CryoSparc[67]. Dose fractionated movie frames were aligned using "patch motion correction," and contrast transfer function (CTF) were estimated using "Patch CTF estimation". Exposures with the estimated ctf fit resolution worse than 5.5 were rejected. Automated particle picking was performed using a blob picker from 300 random exposures. After 2D classification, a few classes were selected for the template-based particle picking from the entire dataset. For KefC WT* protein structure, 7,550,000 particles were subjected to reference-free 2D classification. Selected particles were used for ab initio model building with multiple classes and hetero refinements. Around 300,500 particles were used for non-uniform refinement with no applied symmetry, which resulted in 3D reconstruction with an average estimated resolution of 3.38 Å resolution. A final round of masked local refinement further improved the maps with the gold standard FSC resolution estimation of 3.16 Å. The glutathione complex and D156N variant structures were also refined using a very similar strategy, and the final map's gold standard FSC resolution were estimated to be around 3.18 Å and 2.98 Å, respectively.

## Model building and refinement

Maps were sharpened using Phenix *Auto-sharpen* tool[68] or *Coot*[69]. The KefC homology model was generated using AlphaFold[63] and was placed in the cryo-EM map using the *fit-in map* utility of UCSF Chimera[70]. Iterative model building into the map was performed using *Coot*[69], and structures were refined using Phenix real-space refinement[68]. For the glutathione complex and D156N structures, the KefC WT* structure was used as the starting model. Iterative model building and real space refinements were again performed using *Coot* and Phenix, respectively.

## SSM-based electrophysiology measurements

For making proteoliposomes, *E. coli* polar lipids (Avanti polar lipids) were mixed with POPC (Larodan) in the w/v ratio of 3:1 in chloroform, and a thin dried film was made under vacuum using a rotary evaporator (Heidolph). The obtained lipids were hydrated by vigorous vortexing in a buffer containing 20 mM Tris-MOPS buffer pH 7.8 and 5 mM MgCl$_2$ to a final concentration of 10 mg/ml. The lipid suspension was freeze-thawed 10 times using liquid N$_2$ and stored at −80 °C. Before incorporating protein, liposomes were extruded through a 400 nm filter 15 times. Proteoliposomes were made with the lipid-to-protein ratio (LPR) of 5:1. Briefly, 50 µl of extruded lipids (500 µg) were mixed with 100 µg of purified KefC protein and 0.65% (w/v) of sodium cholate and incubated for 5 min at room temperature. The mixture was then passed through PD-10 desalting column (Cytiva life sciences) pre-equilibrated in 20 mM Tris-MOPS buffer pH 7.8 and 5 mM MgCl$_2$ buffer to remove the detergent. The liposomes were subjected to ultra-centrifugation at 100,000 × $g$ at 4 °C for 1 h. The pelleted proteoliposomes were resuspended in 100 µl of the same buffer to a final lipid concentration of 5 mg/ml, flash frozen in liquid nitrogen, and stored at −80 °C. 5 mm SSM sensors (Nanion) were activated and prepared using the manufacturer's instructions. Proteoliposomes were diluted 5-fold to a final concentration of 1 mg/ml in non-activating buffer containing 20 mM Tris-MOPS pH 7.8, 5 mM MgCl$_2$, and 200 mM NaCl. Diluted proteoliposomes were also freeze-thawed and sonicated for 30 s in bath-Sonicator 2 times and 20 µl of the sample was loaded on the SSM sensors.

SSM-based electrophysiology was performed using a SURFE2R N1 instrument (Nanion Technologies) as described previously[40,50]. Briefly, KefC activation was achieved by fast solution exchange from a non-activating buffer containing 200 mM NaCl/Choline chloride to an activating buffer containing x mM of KCl, and associated transport/binding was measured. Typically, a buffer with 200 mM KCl was mixed with the non-activating 200 mM NaCl/Choline chloride containing buffer to yield an activation buffer with desired (x mM)

KCl concentrations. without affecting the overall ionic strength of the buffer. The proteoliposomes with different LPR ratios (5:1 7.5:1, 25:1 and 50:1) were also prepared using the method described above, and the peak currents were measured with 25 mM KCl in activation buffer and 200 mM choline chloride in the deactivation buffer.

## KefC thermostablization with lipids

KefC WT*-GFP fusion was purified as previously described and dialyzed against buffer containing 20 mM Tris pH 7.5, 300 mM KCl, 1 mM AMP, and 0.03% (w/v) DDM. The purified fusion protein was diluted to a final concentration of 0.01 mg/ml. Stock solutions of DOPG (840475, Avanti Polar), DOPC (850375, Avanti Polar), and DOPE (850725, Avanti Polar) were prepared by solubilizing these lipids in 10% (w/v) DDM to a final concentration of 30 mg/ml overnight at 4 °C with mild agitation. 96 µl of KefC WT*-GFP was aliquoted into duplicate 1.5 ml tubes and 12 µl of lipid was added followed by addition of β-OG to a final concentration of 1% (from a 10%w/v stock). For negative control the KefC WT*-GFP was incubated with 1% DDM and 1% β-OG (10%w/v stock). The samples were heated at 40 °C for 10 min and centrifuged at 5000 g for 30 min. 80 µl of supernatant from each sample was injected onto an Enrich SEC 650 10 ×300 Column (Biorad) preequilibrated in buffer containing 20 mM Tris pH 7.5, 150 mM KCl, 1 mM AMP, and 0.03% (w/v) DDM for fluorescence detection size exclusion chromatography using a Shimadzu HPLC LC-20AD/RF-20A system. The peak heights were normalized with KefC WT* non heated sample as highest point and KefC WT* heated sample as lowest point for two independent fsec runs and further plotted using Graphpad Prism (9.5) software.

## Thermostablization of KefC mutants

Membranes for KefC WT*-GFP fusion and mutants were prepared as mentioned above. The fluorescence from the GFP-fusions was measured with microplate spectrofluorometer (Thermo Scientific Fluorskan) and membranes were solubilzed in buffer containing 25 mM Tris pH 7.5, 300 mM KCl, 1 mM AMP to achieve similar fluorescence intensities in the samples. For AMP free samples, membrane pellets were resuspended and diluted in 25 mM Tris pH 7.5, 300 mM KCl. Non-solubilised membrane material was pelleted by ultracentrifugation at 195,000 × $g$ for 45 min at 4 °C. 96 µl of the supernatant solution was aliquoted in triplicates and β-OG was added to a final concentration of 1% (from a 10%w/v stock) to all samples. The samples were incubated at either at 4 °C or 60 °C for 10 min and centrifuged at 5000 g for 30 min. 80 µl of supernatant from each sample was injected onto an Enrich SEC 650 10 × 300 Column (Bio-Rad) preequilibrated with buffer containing 20 mM Tris pH 7.5, 150 mM KCl, 1 mM AMP and 0.03% (w/v) DDM for fluorescence detection size exclusion chromatography. The peak heights were normalized with respect to highest value of each mutant's non heated sample. The normalized data was plotted using Graphpad Prism (9.5) software.

## Alkaline phosphatase treatment

KefC WT*-GFP and R146A mutant membranes were prepared as mentioned above and solubilized in 25 mM Tris pH 7.5, 300 mM KCl,1 mM AMP buffer. 100 µl of total sample volume in triplicates was prepared with addition of 1 unit of Shrimp Alkaline Phosphatase (Sigma-GEE700927). The samples were incubated for 30 min at 4 °C. For negative control, the Shrimp Alkaline Phosphatase was replaced with equal volume of 20 mM Tris pH 7.5, 300 mM KCl,1 mM AMP buffer. The samples were centrifuged at 5000 × $g$ for 30 min at at 4 °C. 80 µl of supernatant from each sample was injected onto an Enrich SEC 650 10 × 300 Column (Biorad) pre-equilibrated with buffer containing 20 mM Tris pH 7.5, 150 mM KCl, 1 mM AMP and 0.03% (w/v) DDM for fluorescence detection size exclusion chromatography. Data was plotted using Graphpad (9.5) Prism software.

## KefC thermostablization with nucleotides

Membranes of KefC WT*-GFP were solublized in buffer containing 20 mM Tris pH 7.5, 300 mM KCl, 1 mM AMP (A2252 Sigma Aldrich) or 1 mM CMP (C1131 Sigma Aldrich), or 1 mM cyclic di-AMP (17753 Cayman chemicals) for 1 h. Non-solubilised membrane material was pelleted by ultracentrifugation at 195,000 × $g$ for 45 min at 4 °C. 96 μl of the supernatant solution was aliquoted and β-OG was added to a final concentration of 1% (from a 10% w/v stock) to all samples. For negative control, the nucleotide was replaced with equal volume of 20 mM Tris pH 7.5, 300 mM KCl, 0.03% (w/v) DDM buffer. The samples were incubated at either at 4 °C or 40 °C for 10 min and centrifuged at 5000 g for 30 min. 80 μl of supernatant from each sample was injected onto an Enrich SEC 650 10 × 300 Column (Bio-Rad) preequilibrated with buffer containing 20 mM Tris pH 7.5, 150 mM KCl, and 0.03% (w/v) DDM for fluorescence detection size exclusion chromatography.

## Free energy calculation for Na⁺ versus K⁺ ion binding

All simulations were performed using GROMACS 2022.5[71]. The KefC protein was embedded in a pure POPC membrane using CHARMM-GUI membrane bilayer builder[72,]. The system was equilibrated in presence of 150 mM KCl following the CHARMM-GUI protocol[72–74] using the CHARMM36m forcefield[75] and the TIP3 water model[76]; Van de Waals interactions were calculated with a cutoff radius of 1.2 nm, while electrostatic interactions were calculated using the Particle Mesh Ewald method (PME[77]), with a cut-off of 1.2 nm. The minimization was followed by 6 steps of equilibration, in which positional restraints were slowly released. In brief, the first three steps were performed for 125 ps with a time step of 1 fs with positional restraints on the backbone (step1: 4000 kJ, step2: 2000 kJ, step3: 1000 kJ), side chains (2000 kJ, 1000 kJ, 500 kJ), dihedral angles (1000 kJ, 400 kJ, 200 kJ) and lipids (1000 kJ, 400 kJ, 400 kJ); the following three steps were performed for 500 ps with a time step of 2 fs with positional restraints on the backbone (step4: 500 kJ, step5: 200 kJ, step6: 50 kJ), side chains (200 kJ, 50 kJ, 0 kJ), dihedral angles (200 kJ, 100 kJ, 0 kJ) and lipids (200 kJ, 40 kJ, 0 kJ). In the equilibration steps, the Berendsen thermostat[78] was used with a coupling temperature of 303.15 K, and the Berendsen barostat[78] was used with a semi isotropic pressure coupling scheme. Three unrestrained 10 ns simulations were then performed, using the v-rescale thermostat[79] and c-rescale barostat[80]. 100 snapshots were extracted from each simulation and used as starting structures for the alchemical transformation. Switching for each replica was carried out for 50, 100, 200, and 500 ps, where we alchemically mutated K⁺ ions to Na⁺ ions in the protein and in water to extract the relative free energies of binding. The resulting work distributions were analyzed with the Crooks Fluctuation Theorem[81], as implemented in pmx[82].

## KefF cloning and purification

The KefF encoding gene was amplified using primers containing XhoI and XbaI restriction sites (Supplementary Table 1). Genomic DNA present in the cell lysate of *E. coli* MACH1 cells (Thermo Scientific) was used as a template for PCR. Complementary sticky end overhangs were produced following digestion using XhoI and XbaI restriction enzymes (NEB) and ligated into the pET303 vector digested the same way. The resulting digests were ligated using T4 DNA ligase (NEB).

The KefF construct was transformed into *E. coli* Lemo (DE3) strain (NEB) for overexpression. In brief, 4 L of TB media (Formedium) supplemented with 100 μg/ml Ampicillin was inoculated with a saturated primary culture to a final OD₆₀₀ of 0.1 AU. The cultures were grown at 37 °C at 180 rpm until OD₆₀₀ of 0.6 AU was reached. Protein overexpression was induced by addition of IPTG to a final concentration of 0.4 mM, and cell cultures were further incubated for 16 h at 25 °C. *E. coli* cells were harvested by centrifugation at 4000 × $g$ for 10 min, supernatant discarded, and the cell pellet was resuspended in buffer A containing 20 mM Tris pH 7.5, 300 mM NaCl, and 20 μM FMN (Sigma).

Resuspended cells were lysed by mechanical disruption at 30 kPsi by two passes through a Constant Systems cell disruptor. Remaining cells and cell debris were removed by centrifugation at 10,000 × $g$ for 30 min at 4 °C. The supernatant was incubated with 10 ml of Ni-NTA agarose resin (Qiagen) pre-equilibrated with buffer A for 90 min. The Ni-NTA slurry was transferred to a 2.5 × 20 cm Econo-glass column (Bio-Rad) and washed with 250 ml of buffer A 50 mM Imidazole. KefF was eluted with 40 ml elution buffer containing 20 mM Tris pH 7.5, 300 mM NaCl, 20 μM FMN, and 250 mM Imidazole. The eluted KefF was dialyzed overnight at 4 °C against 3 L of 20 mM Tris pH 7.5, 150 mM NaCl and 20 μM FMN in 3.5 kDa MWCO dialysis-tubing (Snake skin, Thermo scientific) and concentrated using a 10-kDa MW cutoff concentrator (Millipore) at 3000 × $g$, 4 °C. The concentrated KefF protein was injected onto an Enrich SEC 650 10 × 300 Column (Bio-Rad) preequilibrated with the dialysis buffer and the main peak collected.

To study KefC and KefF interaction, purified KefC and KefF were concentrated to 65 μM and 50 μM respectively, and mixed to final molar ratios of 20:50, 20:100, and 20:200 μM, into total volumes of 100 μL respectively. Concentrated KefF was also supplemented with 0.03% (w/v) DDM, prior to mixing to prevent any detergent dilution. The samples were incubated on ice for 30 min and 70 μl from each molar ratio was injected onto an Enrich SEC 650 10 × 300 Column (Bio-Rad) preequilibrated with the reaction buffer containing 20 mM Tris pH 7.5, 150 mM KCl, 1 mM AMP and 0.03% (w/v) DDM and size exclusion chromatography was performed. Protein peaks were detected by absorbance at 280 nm

## Reporting summary

Further information on research design is available in the Nature Portfolio Reporting Summary linked to this article.

## Data availability

Data supporting the findings of this paper are available from the corresponding author upon request. Source data is provided with this paper in the Source data file. The coordinates and the maps for KefC, KefC D156N, and KefC with GSH have been deposited in the Protein Data Bank (PDB) and Electron Microscopy Data Bank (EMD) respectively with entries 8BXG, EMD-16318 (KefC); 9EMB, EMD-19816 (KefC Asp156Asn variant); PDB: 8BY2, EMD-16319 (KefC with GSH). Source data are provided with this paper.

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

## Acknowledgements

We are grateful to Magnus Claesson for critical feedback of the manuscript and Marta Carroni at the Cryo-EM Swedish National Facility at SciLife Stockholm for cryo-EM data collection. This work was funded the Swedish Research Council, Göran Gustafsson Foundation, and a European Research Council (ERC) Consolidator Grant EXCHANGE (Grant no. ERC-CoG-820187) to D.D.

## Author contributions

D.D. and A.G. designed the project. Cloning, expression, and purification was done by A.G. and S.K. Sample preparation for cryo-EM and screening was carried out by A.G., P.F.M., and S.K. Cryo-EM data collection was carried out by A.G., A.P.B., and P.F.M. Map reconstruction and model building was carried out by A.G. with help from R.M. SSM-based experiments were carried out by S.K and A.G. The MD simulations were performed by C.A. All authors discussed the results and commented on the manuscript.

## Funding

## Competing interests

The authors declare no competing interests.
