## [Peer Review File · Nature Communications]

Structure and mechanism of the K⁺/H⁺ exchanger KefC

Editorial Note: Parts of this Peer Review File have been redacted as indicated to remove third party material where no permission to publish were obtainedREVIEWER COMMENTS

Reviewer #1 (Remarks to the Author):

Gulati et al present a long-awaited structure of the K⁺/H⁺ antiporter KefC. This by itself is a major breakthrough and very exciting! However, the manuscript itself could be improved on the data interpretation prior publication:

1. Particularly, throughout the manuscripts the aspects regarding potassium homeostasis in bacteria are not very accurate. To my knowledge, KefC is not described to be involved in osmoadaptation, as mentioned several times. It is important for lowering the pH to protect against electrophiles. The use of c-di-AMP (not di-c-AMP) is not described and also not logical. *E. coli* does not have c-di-AMP and neither are homologs described to be regulated by c-di-AMP. The description of TrkA is misleading. TrkA is not a domain of TrkH but an individual, cytosolic protein made of two RCK domains. Four TrkAs, made of in total eight RCK domains, form a gating ring that associates to dimeric (two-pored) TrkH. I would also like to see an alignment that proves that KefC_RCK is the closest related to TrkA rather than to other RCK proteins or domains. In the databases, there is a general misassignment of many RCK domains as TrkA domain, while they have e.g. less sequence identity to TrkA than to MthK_RCK. In many cases, TrkA is just an equivalent of RCK.

2. I also have some comments to the structural models. To validate the reliability of the assignment of densities to K⁺ and AMP, it would be important to see a bit of the surrounding map in Supplementary Figure 3 and/or indicate the sigma used for all sub-maps shown. The maps with K⁺ and AMP fitted look rather noisy so I am wondering how reliable the assignment is. On the GSH-bound map, I am missing detailed information on the local resolution. With respect to the nucleotide binding, it would be of interest to discuss why KefC is specific to AMP and does not (?) bind ADP and ATP, while the structurally similar TrkA (both have a Rossman fold with conserved nucleotide binding motif) binds ADP and ATP and not AMP. Was this actually experimentally tested? Might be more interesting than the shown approaches with c-di-AMP and CMP (Supplementary Figure 1).

3. This said, I would have appreciated some more functional data to better justify the correct assignment. The SSM-based electrophysiology measurements could have been done with further variants mutated in the K⁺ binding site. The mutations tested (D156N, Q155D) were rather drastic and completely abolished activity. Instead, less severe mutation should result in decreased affinities rather than in the completely abolished activity.

4. In general, I am lacking several controls on the SSM-based electrophysiology to judge. What is the orientation of the protein in the liposomes? Do the transient currents really originate from binding only?

To me, they don't look steep enough. Do you perhaps observe a half-cycle and thus determine the K_m rather than the K_d ? Is a half-cycle actually possible or is the protein still inhibited by AMP? To address these questions, different LPRs (the slope should remain identical if only binding is sampled) and different liganded state (with activating GS-X added, with further AMP added or with AMP binding abolished) should be tested? Also, the reflection on the selectivity is incomplete. Does KefC actually bind or transport Rb^+ ?

5. Similarly, the experiments concerning the RCK interactions and their role in inhibition (lines 288-297) should be fleshed out more. A start would be testing a construct in which the RCK domains are deleted, which should be constitutively active. Other potential experiments include changing the length/composition/flexibility of the RCK linkers to modulate the available conformations for the RCK domains. This could also help shed light on the two separate RCK orientations you observe in your structure. Mutating the residues involved in the salt bridges of the transporter/RCK interaction should likewise abolish inhibition and lead to constitutively active transport. Alternatively, stabilizing these interactions should constitutively inactivate the transporter as well as preventing activation by Keff, which, while proving difficult structurally, may be probed with functional assays; this could be achieved by cross-linking, or by switching the salt bridges involved, which may change the electrostatic landscape enough to prevent the RCK-Keff interaction. Finally, showing the association/dissociation of the RCK domain in the absence and presence of AMP/GSH by e.g. FRET or EPR would greatly support the viability of the proposed inhibition model and underscore the role of GSH binding in stabilizing the inhibited, dimerized conformation.

6. Another question that should be addressed in more detail is the dimerization of the RCK domains. What hints can the structure obtained give us about the activation mechanism of KefC? AMP and GSH favor dimerization and, thus, inhibition of the transporter. What happens when the residues involved in the dimerization or AMP binding are mutated? Could AMP be exchanged for an activating nucleotide like in Trk? Was the effect of GSX conjugates on RCK-mediated inhibition tested to support the hypothesized activation mechanism (Lines 365-369)? Do the individual particles of the AMP-dialysed dataset show a signal for the RCK domains? I.e. do they unfold from AMP dialysis or do they monomerize and become so flexible that they can no longer be aligned properly in 2D classes, resulting in no apparent signal? Rather than attempting to solve a structure with AMP removed by dialysis, which appears to destabilize the RCK domains, would a displacement by GSX not be the more promising structural approach?

In addition to these four major points here are some comments I noted while reading the manuscript:

- Line 39: "by cation ion-channels."
- Line 38: "... and that ion-selectivity..." for clarity
- Line 48: Use 'achieve' or 'mediate' rather than 'combat'
- Line 53: remove comma after '(KEA1-6)'

- Line 57: 'functions' plural
- Line 62: add comma after 'electrophiles'
- Line 70: I would suggest a new paragraph focused on the TMD/transport mechanism, starting with the evolutionary relation to NapA, followed by domain structure, and ending with the mechanistic discussion.
- Line 72: add 'a' "...typified by a six transmembrane..."
- Line 74: change to 'evolutionarily'
- Line 88: change 'suspectable' to 'susceptible'
- Line 101: switch 'bacterial' and 'homologous'
- Line 102: "founding member NhaA" the meaning of founding member is unclear.
- Line 106: remove hyphen 'evolutionary relationship'
- Line 107: clarify "...between KefC protomers..."
- Lines 111-112: '...thermostabilization of detergent-purified KefC upon the addition of either DDM and POPC or E. coli lipids POPE and POPG by heat FSEC.'
- Line 115: exchange 'refined' for 'modelled'
- Line 116: exchange 'transverse' (adjective) for 'span'
- Line 133: "...only three residues each in length..." compared to how long in NapA/other Na/H antiporters? The consequence of this "more close-knit crossover" does not immediately become clear, and the concept would be easier introduced in line 178 when its implication in ion selectivity is discussed. It may be interesting to mutate the peptide breaks to more closely mimic the length of the Na⁺/H⁺ antiporters and see if that changes ion selectivity, which would strengthen the structural interpretation. What happens to transport/selectivity when the Ser/Lys interaction near the ion binding site is mutated?
- Line 134: "close-knit"
- Figure 2b: Glutamate 337 is labeled as D337
- Supplementary Figure 1a-d: Instead of/in addition to labeling each chromatogram with 'KefC' consider adding labels that clarify the differences in buffer conditions
- Supplementary Figure 1f: Legend reads thermal stabilization by lipids, but the figure shows stabilization by nucleotides
- Line 175: "...to inward-open- NapA."
- Line 176: Exchange 'to' to 'and'; remove 'either'
- Line 187: May be clearer to write '...forms an additional hydrogen bond with Ser125...'
- Line 199: Replace 'More specifically' with 'In the case of KefC, this is the binding...'

- Line 202: change ‘an’ to ‘a’
- Line 202: change to ‘...a “dead” Asp56Asn variant in which ion binding is precluded...’
- Line 207: change ‘negatively’ to ‘negative’
- Line 240: to clarify, change to ‘...the RCK domains seen in Trk K⁺ channels interact with the TM domains directly via their Rossman fold.’
- Line 241: Note that this observation is true for this structure. Other states could also exist.
- Lines 252-253: Perhaps also comment on the implications of these two different conformations/assemblies in the same structure. Could they represent different regulation states? Are they fixed states at all or do you see indicators for a flexible domain, of which your structure is an average? i.e. is the local resolution of the RCK domains the same? Do the different interactions of the RCK domain result in structural differences in the TMD? Or could they imply concerted/individual control of transport modules by the RCK domains? Is there an argument to be made for/against cooperativity? Does this observation make sense mechanistically in a physiological context? Could it explain why KefC is a dimer (TrkH as a dimeric channel is controlled by a single RCK ring, making the regulation more efficient, for example)? What implications does the apparent flexibility observed in the structure here have for the activation of KefC by KefF?
- Figure 3b: Add a legend in the panel which domains have which color
- Line 278: change ‘to’ to ‘with’
- Line 279: change ‘abolishing’ to ‘abolish’
- Lines 279-286: The section on the 3DVA +/- GSH is difficult to understand without the assumption that the RCK domains regulate by association/disassociation, which is only introduced in the next paragraph. Particularly the sentence “Indeed, the Arg416Ser variant abolishing GSH inhibition results in a constitutionally active transporter” only makes sense in light of the hypothesis that abolishing GSH binding through this mutation subsequently prevents association of the RCK domains and inhibition of the transporter. It would be better understandable if this section were moved to the next paragraph, after this interaction hypothesis has been introduced.
- Line 304: ‘KdpFABC’
- Line 335: change ‘conformations’ to ‘conformational changes’; change ‘substrate bound ion’ to ‘substrate binding site’ or ‘ion binding site’
- Line 369: I would be careful describing the alphafold prediction as a conformational state – particularly for multi-domain proteins, AF has a tendency to predict the individual domains with high accuracy, while moderately flexible to flexible linkers are often predicted in a random orientation with a high uncertainty score, as is the case for the KefC prediction.
- Line 990: Capitalize ‘C’ in ‘KefC’
- In general: Check hyphenation, things like ion transporters (line 20), ion selectivity (line 38), distantly related (line 51) for example need not be hyphenated

Reviewer #2 (Remarks to the Author):

The manuscript by Gulati et al. addresses Kef transporters belonging to the CPA2 superfamily. Kef transporters are K⁺/H⁺ antiporters and are part of the diverse biological response bacteria have to maintain potassium homeostasis.

The manuscript presents two cryo-EM structures of the bacterial K⁺/H⁺ antiporter KefC. In addition, the purified KefC protein in complex with AMP has been reconstituted into proteoliposomes and used for Solid Supported Membrane (SSM) based electrophysiology. Finally, the manuscript presents some MD to support the identification of a spherical density in the expected binding site as potassium.

Several findings in the manuscript are of interest to the community. This represents the first structure of a Kef K/H transporter. KefC has a C-terminal RCK domain and the manuscript presents the first glimpse of how this regulatory domain interacts with the transporter domain. In addition, the structures reveal a spherical density that is very likely a potassium ion, and thus represent the first substrate bound structure from the CPA2 superfamily.

The observation that a central crossover in the transmembrane part is shorter than in other CPA2 members and how this related to substrate recognition and specificity is very interesting.

For me, the key point of the manuscript is the regulation/inhibition of a CPA2 transporter by a RCK domain, very similar to the regulatory domains found on potassium channels such as TrkH. This is best exemplified by the AMP/GSH complex structure.

The structural data is convincing. The SSM data less so in its present form. I am not an MD expert and will refrain from commenting on that except to say that it would be beneficial to use a bit of time to better describe the aim and methodology of this part for the non-expert (e.g. at line 181). The methods are well described in general otherwise.

Major points:

Is the AMP-only sample an inhibited state or an active state? This is not discussed anywhere and is a major omission.

Briefly at least four states are implied by the study as I understand it:

1) AMP-only bound: The first structure presented and the sample used for SSM. Is this sample active or inactive? A comparison (superposition, RMSD) to the AMP/GSH structure is completely lacking and this should be rectified. Presumably this AMP-only form is not physiologically relevant (based on the intro and fig 1a)?

2) AMP and GSH bound (biologically relevant): This is actually the most interesting structure in the manuscript but is only mentioned at the end. This should be the main finding of the paper in my opinion, as we now have a biologically relevant and well-defined inhibited structure.

3) GS-X only bound: This is presented as the physiologically active form. Note that fig 1a (incorrectly?) show that GS-X and AMP bind at the same time. This is at odds with e.g. line 367.

4) apo-form, nothing bound: This is made in the study by dialyzing away the AMP. This sample is not physiologically relevant but is expected(?) to mimic the GS-X active form, based on the observed mobility of the RCK domains.

The SSM is done using the AMP-only sample. Assuming that this is similar to the AMP/GSH structure it should be inactive. Yet SSM shows a signal. Do the authors believe that the signal is from the half-reaction (as implied, e.g. line 199-201) or from the binding of K⁺ to the inhibited protein? This need to be clearly addressed and explained and related to the expected mode of inhibition (presumably inhibition is by locking the transporter in the inward state?).

Is the domain swap of the RCK domains a relevant physiological feature or a sample artifact? This should be discussed in the manuscript.

The fact that the two RCK domains are not binding the transporter monomers in a symmetrical fashion is very interesting but is not discussed. Why is binding asymmetrical? More effort should be made to explain how inhibition is achieved in general. Do you have any clues from the structure. Do you think the monomers can transport K⁺ individually or is transport in the dimer coupled. Do you think that both interactions from the RCK domains to the transporter domains lead to inhibition, or is only one monomer inhibited in the presented state?

SSM-data:

The statement Line 203-206 seems confused or perhaps even wrong? Electroneutral transport cannot be distinguished from electrogenic transport by the directionality of the peak. The directionality shows if a positive change is going into or out of the proteoliposome. E.g if transporting a negative substrate, the current would be negative.

L201: Ph 7.8 is mentioned vs fig 2d legend that mention pH 8.5?

L204: "in the presence of 200 mM NaCl" vs the figure legend where 30mM KCl is mentioned? There are many similar omissions and inconsistencies in the text and figure legends.

I agree that the current peak observed likely represents binding and not transport. However, this is not demonstrated, and could readily be done by measuring the full peak width at half maximum (FWHM) using different Lipid to Protein Ratios (LPR). If we only observe binding, then FWHM should be independent of LPR. If transport is involved, the FWHM will increase with increasing LPR. For an example of this type of analysis see for instance figure 1 in Bazzone, A., Zabadne, A.J., Salisowski, A., Madej, M.G., and Fendler, K. (2017). A loose relationship: Incomplete H⁺/sugar coupling in the MFS sugar transporter GlcP. *Biophysical Journal* 113, 2736-2749.

Some suggestions. Have you tried this:

Try the apo-form for SSM (i.e.. dialyze away the AMP). What happens?

If you think the SSM sample is inactive, add GS-X to activate. What happens?

If you believe the SSM sample is active, add GSH to inhibit. What happens?

L282: Arg416Ser is constitutively active. This mutant should be tested with SSM. What happens?

Could you make a truncated KefC without the RCK domain and test with SSM?

In Fig2d the D156N and 'empty' traces are similar and negative. In sfig6 the empty trace is missing, but the D156N peak has reversed its direction and is now positive and similar to the wt. Explain this observation. Also, the empty trace must be added here. Are you sure the D156N mutant doesn't also bind K⁺ to some extent?

Comparing sfig6a and sfig6c: sfig6c is used to demonstrate that the mutant and wt are indifferent to other ions. without including K⁺ in the same analysis (on the same sensor) this is meaningless. In all cases of sfig6c the current at 25 mM Ion is around 500 pA. This is perfectly comparable to the peak current of K⁺ shown in sfig6a. At present it seems that the KefC is binding or transporting all of these ions regardless of the D156N mutation or not. Showing (from the same sensor) that K⁺ is the only case where a clear difference can be seen between mutant and wt, and also that K⁺ elicits a higher current response than the other ions is a necessary control.

Minor Points:

The manuscript was somewhat difficult to read and has a range of unclear and confusing statements that should be addressed. Some examples:

L93: active pH. What does this mean?

L94: "images" This is micrographs or movies.

L138: K307 is on TM10, but in fig 2b it is on TM11?

L139-145. The intermittent discussion of specific NapA residues here and elsewhere makes the text hard to read. Perhaps move to a discussion later?

For both cryo-EM datasets, very large datasets were collected (up to 25K movies and 7-10M particles picked), but in the end only ~300K particles were used for the reconstruction. It would be interesting to discuss this a bit. Is it possible that other conformations of KefC (e.g. symmetrical conformations) were present in the data for instance?

Title: Definition vary in the literature, so feel free to ignore, but the word exchanges should be reserved for transporters that exchange compounds (e.g ADP/ATP exchanger or the CMP/CMP-sialic acid

exchanger), while secondary active transporters that are driven by sodium or proton gradients are antiporters (or symporters depending). In the manuscript the word exchanger is only used in the title, not the main text where antiporter or transporter is used.

Abstract: Only one structure is mentioned but two are presented.

Intro: be clear that K⁺ transport is driven by proton transport. Discuss references on stoichiometry. What is known about this for Kef systems?

Summarize the known structures of RCK domains. Do we have structures with AMP and/or GSH and GS-X bound. This is unclear, but seems very relevant.

L53, 55, 56, 75 and elsewhere. When comparing to other protein families and proteins it would be instructive to have the sequence identify mentioned somewhere. At least for a few cases.

L114: The lipid identity is not super relevant for the findings, and the whole section could be shortened. The identity is speculative based on thermostability only. IS POPG found in E coli membranes, and is it abundant?

L123 An RMSD(Ca) of 4.3 Å seems very high. This number needs to be better explained. In fig 1e it doesn't look like a 4.3 Å RMSD deviation. Maybe our methods to calculate RMSD differ?

L150: you could discuss if the coordinating residues are conserved? I assume so. Does the observed coordination fit with potassium. Related in fig 2c it is impossible to judge the density. What is the local resolution?

L229, DALI hit and "most similar": Everything is most similar to something else. What is the Z-score? What is the RMSD?

L237: compare the structure of the RCK domain to solved structures. It seems like a strange omission to not do this?

L352-360: NHE regulation does not seem that relevant and could be shortened. Maybe it would be more relevant to compare to the TrkH regulation from similar regulatory domains. Do these domains work in a similar fashion?

L384: Since you have KefF purified, try to add it to SSM to see if this improves/changes results?

Table 1: cryo-EM GSH model: Out of curiosity, why did you do manual map sharpening for the AMP/GSH dataset?

fig 1a: This panel could be improved. e.g. arrows showing K and H transport are not clear. Is AMP always present, also with GS-X? From the text it seems like this is not the case.

fig 1e: Labels are needed to orient the reader.

fig 2b: label TM6 also.

fig 2c: It is very hard to see what is happening here with the density. Also label key helices.

fig 2e: no units on Kd value.

fig 3d: is very hard to read. Try to link it to panel c perhaps?

fig 2a & 3e: you should to define the blue-to red range of your electrostatic potential calculations (in kTe-1).

fig 4: Again, somewhat confusing. E.g. only one AMP is shown but each monomer binds AMP. Also, now AMP is gone from the GS-X activated state (in contrast to fig 1a).

Sfig2: Only one curve is shown for the FSC. Which one? You should preferably present “no mask”, “spherical”, “Loose” “tight” and “corrected”. For a discussion see e.g. Chen, S. et al. High-resolution noise substitution to measure overfitting and validate resolution in 3D structure determination by single particle electron cryomicroscopy. *Ultramicroscopy* 135, 24–35 (2013).

Reviewer #3 (Remarks to the Author):

The authors report on the structure determination of the K⁺/H⁺ transporter KefC through cryo-electron microscopy. The mechanism of transport is also inferred by combining structural data and other experimental techniques, including electrophysiology. The study is complemented with Molecular Dynamics simulations aimed at elucidating the determinants of selectivity for K⁺ versus Na⁺.

The paper is very clearly written, and in my opinion, is an excellent piece of work overall.

Unfortunately, concerning my field of expertise, I have some concerns regarding how the free energy calculations have been performed.

Specifically, I am rather confused about whether the authors performed the K⁺->Na⁺ transformation using equilibrium or out-of-equilibrium simulations. I am pretty sure that the plots reported in Figure S5 are work profiles, but the authors employ terminology that is consistent with equilibrium simulations, like “FEP” and “windows”. In particular, term windows should be reserved for staged calculations along the perturbation. Instead, to my best understanding, the authors extracted 100 configurations and for each of which they performed an out-of-equilibrium switching, am I right?

Also, it is not clear which estimator the authors employed, as even though they refer to the Bennet Acceptance Ratio, the plots shown in Figure S5 rather suggest that the work distributions were used according to Crooks’ theorem.

I would strongly advise the authors to provide some clarification of their computational setup. Please notice that this is not (only) a matter of being picky regarding the proper terminology, but it is also related to the reliability of results. Indeed, if the free energy difference was estimated using out-of-equilibrium techniques, 50 ps of simulation time might be enough to obtain meaningful results (with a huge number of realizations), but in my experience, these simulations are way too short in the case of equilibrium runs.

In any case, authors are encouraged to increase the length of their simulations to provide an indication that the estimated free energy difference is close to convergence.

Apart from that, the free energy difference for the K⁺->Na⁺ transformation in water should be closely related to the solvation-free energy difference between the two ions. Are results consistent with previous estimates?

Maybe it’s me, but how does the reported free energy difference between the two ions of 15.17 kJ/mol compares with the values reported in Figure S5? (-139.90 and -170.23)

Please provide more details regarding the simulation setup, like the water model, cutoffs employed, long-ranged electrostatics, timestep, and so on. This is very important information for ensuring the reproducibility of results.

We thank the referees for their considered evaluation. We appreciate your patience with our earlier submission, which was partly rushed due to certain reasons. We have taken the time to modify the paper and respond as appropriate to all comments below.

REVIEWER COMMENTS

Reviewer #1 (Remarks to the Author):

Gulati et al present a long-awaited structure of the K^+/H^+ antiporter KefC. This by itself is a major breakthrough and very exciting! However, the manuscript itself could be improved on the data interpretation prior publication:

1. Particularly, throughout the manuscripts the aspects regarding potassium homeostasis in bacteria are not very accurate. To my knowledge, KefC is not described to be involved in osmoadaptation, as mentioned several times. It is important for lowering the pH to protect against electrophiles. The use of c-di-AMP (not di-c-AMP) is not described and also not logical. *E. coli* does not have c-di-AMP and neither are homologs described to be regulated by c-di-AMP. The description of TrkA is misleading. TrkA is not a domain of TrkH but an individual, cytosolic protein made of two RCK domains. Four TrkAs, made of in total eight RCK domains, form a gating ring that associates to dimeric (two-pored) TrkH. I would also like to see an alignment that proves that KefC_RCK is the closest related to TrkA rather than to other RCK proteins or domains. In the databases, there is a general misassignment of many RCK domains as TrkA domain, while they have e.g. less sequence identity to TrkA than to MthK_RCK. In many cases, TrkA is just an equivalent of RCK.

Thank you for raising a number of errors in our manuscript – we should have fixed these prior to submission.

Since this is the first structure of a K^+/H^+ exchanger we wished to convey in the introduction the role of K^+/H^+ exchangers in all organisms. In plants, they have been reported to have roles in osmoregulation and pH, which is typically the role of Na^+/H^+ exchangers, but may relate to the fact the plants have a high concentration of K^+ . In bacteria, they appear to have a niche role in lowering the pH to protect against electrophiles. We have edited the introduction to make the distinction clearer and that KefC in bacteria not have a role in osmoregulation.

We have fixed the typo di-c-AMP, which was naturally meant to be c-di-AMP. We thought it may bind to KefC given the role of the c-di-AMP pathway for K^+ sensing in bacteria (PNAS, 118 (14)), but this doesn't seem to be the case and we were didn't realize this signaling pathway is not used in *E. coli* anyway.

We have edited the manuscript to make it clearer that TrkA is a separate protein. Using the KefC RCK domains as input for Foldseek analysis, the closest non-KefC related hits are K^+ channels with RCK domains, e.g., MthK, KtrC and TrkA protein.

The sequence identity to MthK from *Methananothermobacter thermautotrophicus* is 16.7%, TrkA from *Vibrio parahaemolyticus* is 16.8% and KtrA from *Bacillus Subtilis* is 15.7%. We have now included a sequence and structural alignment of these RCK domains in the paper. We have updated the manuscript to state that the structural similarity is not only to TrkA. The largest difference is the additional C-terminal helix, which has not been observed in structures of MthK, KtrB and TrkA (now labelled $\alpha 7$). Incidentally the last helices of the RCK domain make the majority of interactions to the KefC transporter domains and is required for GSH coordination.

Superimposition of RCK domains from KEFC_ECOLI (PDB: 8BY2) (lightpink) and TRKA_VIBPA (PDB: 4J9V) (Pale green) with rmsd of 1.4, KTRA_BASCU (PDB: 4J91) (Light orange) with rmsd of 1.4, MTHK_METTH (PDB: 6OLY) (light blue) with rmsd of 2.4

2. I also have some comments to the structural models. To validate the reliability of the assignment of densities to K^+ and AMP, it would be important to see a bit of the surrounding map in Supplementary Figure 3 and/or indicate the sigma used for all sub-maps shown. The maps with K^+ and AMP fitted look rather noisy so I am wondering how reliable the assignment is. On the GSH-bound map, I am missing detailed information on the local resolution. With respect to the nucleotide binding, it would be of interest to discuss why KefC is specific to AMP and does not (?) bind ADP and ATP, while the structurally similar TrkA (both have a Rossman fold with conserved nucleotide binding motif) binds ADP and ATP and not AMP. Was this actually experimentally tested? Might be more interesting than the shown approaches with c-di-AMP and CMP (Supplementary Figure 1).

These are all good points and in hindsight we should have better explained the extensive characterization of the RCK domains from KefC that has already been carried out previously.

You raise a number of points here and we have broken them down to the following:

K⁺ binding site

We have been working with Na^+/H^+ exchangers for a number of years and despite our best efforts with high-resolution crystal structures and cryo EM structures, no one has ever seen any non-protein density in the ion-binding site.

Map density of K⁺ binding site and surrounding residues (12Å radii) at the same sigma level of 7.2

We included map density of the side-chains in the main figure (Fig. 2c) together with the map density of the K⁺ ion at the same sigma threshold, which we thought was convincing to show that the map for the side-chains were at a similar level as the ion; sigma levels are a bit arbitrary and we thought this was a better way to support this.

Nevertheless, given the importance of describing the “first” ion-bound state for the large Cation:Proton Antiporter family, we wanted to make this data as robust as possible and a “clean” experiment to validate this. For this reason, we further determined the Cryo-EM structure of the D156N mutant at higher salt concentration (300 mM vs 150 mM KCl) as the D156N mutant does not bind K⁺. The Cryo-EM maps of the D156N mutant were reconstructed to a final resolution of 2.98 Å. Overlaying the maps show the same high-quality of density for the side-chains, but we see no signal for the K⁺ ion in the D156N Cryo-EM maps. We have also added an additional supplementary figure showing density for residues present around the K⁺ binding site.

Orange: KefC WT*
Blue: D156N variant
Both densities were contoured at sigma 7.2 in pymol

RCK domains

Previous biophysical and biochemical studies have been carried out on the RCK domains from *E. coli* and *Shewanella denitrificans* as reviewed recently (Membranes 2023; 13(5), 465).

1. The RCK domains purified from *E. coli* natively retain AMP
2. AMP is required for dimerization
3. ATP does not bind, but ADP might bind weakly
4. GSH does not add in stability of the RCK domains, but mutations near the GSH binding site results in a constitutively active protein.

We have now included a structural overlay of the RCK domains in the cryo-EM structure of KefC with the crystal structure of the RCK dimer in complex with AMP and GSH. We see no obvious differences in their coordination and the RMSD is less than 1Å. We have included more detailed maps in the supplementary figure 4 as pasted here below

Map density for AMP (orange) bound to the RCK domains in the Cryo-EM structure of KefC.

Structural comparison of KefC RCK domains in the cryo EM structure (pink) with the previously determined crystal structures of the soluble RCK-domain (5NC8; blue and 3EYW; cyan).

It is plausible that ADP/ATP does not bind as it would clash with the bound GSH that is positioned only 5.7 Å from the terminal phosphate of AMP... taken from the analysis of the KefC RCK domains they write **"We did not detect any protein with NAD⁺, NADH, or ATP bound, which implies that in the cell the domain has a binding constant for AMP at least 200-fold tighter than that for NAD⁺, 6-fold tighter than that for NADH, 700-fold tighter than that for ATP, and 35-fold tighter than that for ADP (ref Biochemistry, 2017 Aug 15; 56(32): 4219–4234.)"**

KefCWT Cryo-EM structure with AMP only (pink), AMP and GSH (sand) and crystal structure of soluble RCK domain with AMP and GSH (3L9W)*

3. This said, I would have appreciated some more functional data to better justify the correct assignment. The SSM-based electrophysiology measurements could have been done with further variants mutated in the K⁺ binding site. The mutations tested (D156N, Q155D) were rather drastic and completely abolished activity. Instead, less severe mutation should result in decreased affinities rather than in the completely abolished activity.

We wanted to convey that the specificity for K⁺ was not as simple as a single residue difference, but the whole architecture is important. We were surprised by both the architecture of the cross-over and the different interactions in the ion-binding site. We have now included a T127V mutant that was a bit more subtle as this residue is valine in NapA and we hoped that we could start to see Na⁺ binding. However, the T127V variant abolished K⁺ binding and did not show any Na⁺ binding. Nevertheless, we think the new structure of the D156N mutant means that we can be more confident in the ion-coordination and, as expected, switching ion-specificity is likely to be more complex than substituting a few side-chains.

Left: Fit of the transient currents as a function of K^+ concentrations for KefC WT. No clear binding affinity could be calculated for KefC D156N (orange), Q155D (green) and T127V (blue) variants. Right: Fit of the transient currents as a function of LiCl and NaCl concentration for KefC T127V variant. Error bars are the mean values \pm s.d. of $n=3$ titrations (sensors) that were each measured in triplicate.

4. In general, I am lacking several controls on the SSM-based electrophysiology to judge. What is the orientation of the protein in the liposomes? Do the transient currents really originate from binding only? To me, they don't look steep enough. Do you perhaps observe a half-cycle and thus determine the K_m rather than the K_d ? Is a half-cycle actually possible or is the protein still inhibited by AMP? To address these questions, different LPRs (the slope should remain identical if only binding is sampled) and different liganded state (with activating GS-X added, with further AMP added or with AMP binding abolished) should be tested? Also, the reflection on the selectivity is incomplete. Does KefC actually bind or transport Rb^+ ?

To clarify in electroneutral Na^+/H^+ exchangers we can only measure the half-cycle (the first K^+ -translocation event) and, as such, we think it is more appropriate to refer to this as binding (K_d) as we discussed previously (NSMB 29:108–120 (2022)). In hindsight we should, however, have included the LPR ratios in the initial submission. We see that the normalized peak currents produced no systematic differences in the calculated time constant from the calculated decay currents. The transported charge should increase current decay towards lower LPRs. The SSM-data is most consistent with binding rather than ion transport, which would be with our preparation capturing a mostly inactivated state.

Transient currents for KefC-WT transporter reconstituted at different lipid-to-protein ratios (LPRs). The transient currents were recorded for LPR ratios 5 (black traces), 7.5 (yellow traces), 25 (blue traces), 50 (pink traces) and empty liposomes (green) are shown above. The peak currents were normalized and decay constants were calculated for $n=3$ different titrations on each independent sensor. Based on the decay constant (τ) values are similar we can conclude that the peak currents are most consistent with a binding event rather than actual transport.

5. Similarly, the experiments concerning the RCK interactions and their role in inhibition (lines 288-297) should be fleshed out more. A start would be testing a construct in which the RCK domains are deleted, which should be constitutively active. Other potential experiments include changing the length/composition/flexibility of the RCK linkers to modulate the available conformations for the RCK domains. This could also help shed light on the two separate RCK orientations you observe in your structure. Mutating the residues involved in the salt bridges of the transporter/RCK interaction should likewise abolish inhibition and lead to constitutively active transport. Alternatively, stabilizing these interactions should constitutively inactivate the transporter as well as preventing activation by KefF, which, while proving difficult structurally, may be probed with functional assays; this could be achieved by cross-linking, or by switching the salt bridges involved, which may change the electrostatic landscape enough to prevent the RCK-KefF interaction. Finally, showing the association/dissociation of the RCK domain in the absence and presence of AMP/GSH by e.g. FRET or EPR would greatly support the viability of the proposed inhibition

model and underscore the role of GSH binding in stabilizing the inhibited, dimerized conformation.

These are all good suggestions and we had, in fact, done a number of them. We first made C-terminal deletions of the RCK domains as we thought this should be constitutively active. Unfortunately, the truncated protein constructs (KefC- Δ 399 and KefC- Δ 379) are prone to aggregation and therefore we cannot isolate the protein. It seems the protein is not very stable when the RCK domains are not present. In addition when the protein is heated without supplementation of additional AMP the protein is predominantly in its monomeric form (Supplementary 11c). It is not clear to us why this is the case, since the related bacterial Na^+/H^+ exchangers are stable homodimers without any cytosolic domains.

SEC profiles of the purified KefC- Δ 399 and Δ 379 variant. Removal of RCK domain clearly led to instability and aggregation in detergent.

Our hypothesis was that AMP influences KefC stability because it was required for RCK dimerization and the RCK domains further interacted with the transporter module. We previously demonstrated this by heating KefC in detergent in the presence of AMP and other non-binding nucleotides. Only KefC heated with AMP was able to retain a homodimer, indicating that the protein was indeed less stable without AMP. To directly demonstrate the relationship between AMP-mediated RCK dimerization and transporter interaction, we have now incubated KefC membranes with shrimp alkaline phosphatase for 30 mins at 4°C (enzyme removes the terminal PO_4^{2-}). Remarkably, the KefC protein dissociates into monomers. This is consistent with the empirical observation that we needed to add AMP to purify stable KefC homodimers.

We have further made a considerable number of mutations for residues making interactions between the RCK domains and the KefC transporter module. We have then compared the propensity of these variants to stabilize the KefC homodimer in the presence of AMP. Compared to the construct of KefC used for structural work, some variants (E465A, R543A, H259) show poorer stability to such an extent that the homodimer can no longer be fully retained during heating at 60°C for 10 mins (shown above). We interpret this as the RCK domains are no longer able to interact tightly to the protein. Other variants show greater stability than WT and retain a substantial fraction of the homodimer after heating at 60°C. It thus appears that these variants are able to interact more tightly to the KefC transporter. We can confirm that AMP is still

bound in such variants (R146A) as incubating with shrimp alkaline phosphatase still dissociates KefC R146A into monomers.

FSEC traces of DDM solubilized membranes of KefC R146A (black) and after incubation with alkaline phosphatase (orange) at 4°C

Unfortunately, we have been unable to optimize an *in vitro* proteoliposome transport assay for KefC despite considerable attempts to probe the relationship between RCK detachment and KefC activity (There is no published report of KefC activity using purified components). Previously, in specialized K^+ - transport deficient *E. coli* deletion strains, Ian Booth and colleagues have shown that robust activity of KefC requires the auxiliary protein KefF, which is located in the same operon as KefC. In these experiments the amount of K^+ content in cells was measured by atomic absorption spectroscopy.

J Bacteriol. 2000 Nov; 182(22): 6536–6540. [REDACTED]

We have purified KefF, but we have been unable to obtain complex with KefC and we have been unable to see an effect in proteoliposome assays (with pyranine).

SEC traces of purified KefC and KefF complex

There is a crystal structure of KefF in complex with the RCK domain of KefC (PNAS 107 (46) 19784-19789). Because these proteins did not form a complex in solution this was achieved by cloning, expressing and purifying a RCK domain-linker-KefF fusion construct. In this crystal structure, the KefF protein formed a reasonable complex with the RCK domain. However, in the Cryo-EM KefC structure the interface interacting with the KefF protein is unavailable, as it is interacting with the transporter domains. While this may seem like a crystallization artefact, we find it interesting that AlphaFold predicts with reasonable confidence a flipped RCK domain conformation, which would make it accessible to the KefF protein.

Taken together, it is possible that the activated state of KefC required the KefF protein to stabilize a detached state of the RCK domains. We have now updated the manuscript showing this plausible activation mechanism. Since we are unable to obtain this complex in solution, it is unclear if GSX is required to help stabilize the flipped state. Unfortunately, these compounds are not commercially available and we are currently trying to synthesize them. We will continue to attempt to obtain a transport assay for KefC so that we can establish these mechanistic details in a follow-up study.

5. Another question that should be addressed in more detail is the dimerization of the RCK domains. What hints can the structure obtained give us about the activation mechanism of KefC? AMP and GSH favor dimerization and, thus, inhibition of the transporter. What happens when the residues involved in the dimerization or AMP binding are mutated? Could AMP be exchanged for an activating nucleotide like in Trk? Was the effect of GSX conjugates on RCK-mediated inhibition tested to support the hypothesized activation mechanism (Lines365-369)? Do the individual particles of the AMP-dialysed dataset show a signal for the RCK domains? I.e. do they unfold from AMP dialysis or do they monomerize and become so flexible that they can no longer be aligned properly in 2D classes, resulting in no apparent signal? Rather than attempting to solve a structure with AMP removed by dialysis, which appears to destabilize the RCK domains, would a displacement by GSX not be the more promising structural approach?

All very good points. I think our answer to the previous questions clarifies the relationship we shown between AMP and RCK dimerization and RCK dimerization and KefC transporter oligomerization. What we haven't been able to show is what interactions are the critical ones for KefC transporter inactivation, but given we see so much variability in the 3DVA indicates it may just be based on proximity.

We haven't mutated residues involved in AMP binding and dimerization, but we think the experiment with Shrimp Alkaline phosphatase addition (to remove AMP) is a cleaner experiment than mutations to show how AMP is required for an interaction with the KefC transporter.

Previously we missed that residues required for AMP coordination have, in fact, been shown to abolish or severely reduced K⁺ transport in whole cell assays (Biochemistry. 2017 Aug 15; 56(32): 4219–4234). Furthermore, in the same paper they could show that there was cooperative stability upon AMP and GSX addition. Taken together, rather than activation requiring AMP removal by GSX addition as we previously suggested, it seems that the active state still has AMP present and is therefore the flipped RCK model for activation seems the most plausible.

DSF experiment to determine the effect of AMP and GSX on SdKefQCTD
(Biochemistry. 2017 Aug 15; 56(32): 4219–4234)

[REDACTED]

We don't see any aggregation in our sample before blotting that indicates that the RCK domains have unfolded upon loss of AMP, but rather in the 2D classes we have not been able to detect the RCK domains, and that they must be too dynamic and therefore can no longer be aligned.

We have had difficulty obtaining GSX from commercial sources and the several groups we reached out to were unable to help. We will continue with trying to synthesize GSX in-house, but at the present stage we don't have this compound. Given the poor KefC stability without AMP, the KefC-GSH efflux system has been more challenging to study *in vitro* than we had anticipated. We agree that it is interesting to uncover how bacteria have evolved finely-tuned sensing system connected to K⁺ homeostasis, but putting such pathways together takes time. Nevertheless, we think we have learned a lot from the first structure of a K⁺/H⁺ exchanger as a representative member that is spread across all Kingdoms in terms of ion-specificity as well as a working model for its activation.

In addition to these four major points here are some comments I noted while reading the manuscript:

- Line 39: "by cation ion-channels."
Corrected
- Line 38: "... and that ion-selectivity..." for clarity
Corrected
- Line 48: Use 'achieve' or 'mediate' rather than 'combat'
The paragraph has been extensively modified
- Line 53: remove comma after '(KEA1-6)'
Corrected
- Line 57: 'functions' plural
Modified
- Line 62: add comma after 'electrophiles'
Added
- Line 70: I would suggest a new paragraph focused on the TMD/transport mechanism, starting with the evolutionary relation to NapA, followed by domain structure, and ending with the mechanistic discussion.
That works better, thanks!
- Line 72: add 'a' "...typified by a six transmembrane..."
Added
- Line 74: change to 'evolutionarily'
Changed
- Line 88: change 'suspectable' to 'susceptible'
Corrected
- Line 101: switch 'bacterial' and 'homologous'
Changed
- Line 102: "founding member NhaA" the meaning of founding member is unclear.
Suitably modified
- Line 106: remove hyphen 'evolutionary relationship'
Removed
- Line 107: clarify "...between KefC protomers..."

Done

- Lines 111-112: ‘...thermostabilization of detergent-purified KefC upon the addition of either DDM and POPC or E. coli lipids POPE and POPG by heat FSEC.’

Rephrased

- Line 115: exchange ‘refined’ for ‘modelled’

Exchanged

- Line 116: exchange ‘transverse’ (adjective) for ‘span’

Exchanged

- Line 133: “...only three residues each in length...” compared to how long in NapA/other Na/H antiporters? The consequence of this “more close-knit crossover” does not immediately become clear, and the concept would be easier introduced in line 178 when its implication in ion selectivity is discussed. It may be interesting to mutate the peptide breaks to more closely mimic the length of the Na⁺/H⁺ antiporters and see if that changes ion selectivity, which would strengthen the structural interpretation. What happens to transport/selectivity when the Ser/Lys interaction near the ion binding site is mutated?

Good points. We have included the comparison to other representative Na⁺/H⁺ exchangers that all have longer extended peptide breaks. In all these members the TM10/TM11 lysine residue interacts with either a glutamate residue close to the ion-binding aspartates or directly to one of the ion-binding aspartates. It’s completely unexpected that the lysine interacts directly to the peptide break, which is itself via Ser125 contributing to ion-coordination. Supporting the important role of the Lys307 its mutation to alanine caused large destabilization and we were unable to purify this mutant; in contrast we have been able to purify even drastic Lys-to-Glu mutations in NapA (PNAS 114: E1101-E1110).

- Line 134: “close-knit”

Corrected

- Figure 2b: Glutamate 337 is labeled as D337

Corrected

- Supplementary Figure 1a-d: Instead of/in addition to labeling each chromatogram with ‘KefC’ consider adding labels that clarify the differences in buffer conditions

Modified

- Supplementary Figure 1f: Legend reads thermal stabilization by lipids, but the figure shows stabilization by nucleotides

Modified, thanks!

- Line 175: "...to inward-open→ NapA."

Corrected

- Line 176: Exchange 'to' to 'and'; remove 'either'

Changed

- Line 187: May be clearer to write '...forms an additional hydrogen bond with Ser125...'

Modified

- Line 199: Replace 'More specifically' with 'In the case of KefC, this is the binding...'

Modified

- Line 202: change 'an' to 'a'

Changed

- Line 202: change to '...a "dead" Asp56Asn variant in which ion binding is precluded...'

Modified

- Line 207: change 'negatively' to 'negative'

Modified

- Line 240: to clarify, change to '...the RCK domains seen in Trk K⁺ channels interact with the TM domains directly via their Rossman fold.'

The statement is now been modified

- Line 241: Note that this observation is true for this structure. Other states could also exist.

Noted

- Lines 252-253: Perhaps also comment on the implications of these two different conformations/assemblies in the same structure. Could they represent different regulation states? Are they fixed states at all or do you see indicators for a flexible domain, of which your structure is an average? i.e. is the local resolution of the RCK domains the same? Do the different interactions of the RCK domain result in structural differences in the TMD? Or could they imply concerted/individual control of transport modules by the RCK domains? Is there an argument to be made for/against cooperativity? Does this observation make sense mechanistically in a physiological context? Could it explain why KefC is a dimer (TrkH as a dimeric channel is controlled by a single RCK ring, making the regulation more efficient, for example)? What implications does the apparent flexibility observed in the structure here have for the activation of KefC by KefF?

As now clarified, AMP is required for dimerization of the RCK domains and this dimer is sufficient to act in an autoinhibitory manner, by forming extensive interactions to the transporter module. The RCK domains are unlikely to drive a conformational change in the transporter itself, but rather their detachment removes a break so the transporter can begin cycling – a similar autoinhibition mechanism seems apparent in the voltage-gated Na⁺/H⁺ exchanger SLC9C1 (Nature 623:193–201).

Although the RCK domains are asymmetric, it appears to be a fairly stable conformation as observed by the same structural state for three different preparations

of KefC (WT+AMP, WT+AMP+GSH and D156N), as well as the similar local resolution estimates of the RCK domains as the transporter domain. Nevertheless, the RCK domains are somewhat flexible, which can be best seen by 3DVA of the reconstruction of WT+AMP+GSH. Notably, states with more prominent interactions between the $\alpha 7$ in the RCK domain and TM7 in the transporter domain, show stronger map density for GSH (Supplementary Video 1).

The structure of KefC with either AMP or AMP-GSH are very similar (Supplementary Fig. 10c). We have included the 3DVA of the KefC with AMP and, in comparison, the 3DVA with GSH shows states with marginally closer interactions with the transporter domain, but it seems clear that both represent an inhibited state. The flexibility is due to the connecting linker between TM13 and the RCK domains (that we cannot model) and presumably it is an important feature so that the domains can be detached (presumably stabilized by KefF as mentioned).

- Figure 3b: Add a legend in the panel which domains have which color

Has been added

- Line 278: change 'to' to 'with'

changed

- Line 279: change 'abolishing' to 'abolish'

Sentence modified during revision

- Lines 279-286: The section on the 3DVA +/- GSH is difficult to understand without the assumption that the RCK domains regulate by association/disassociation, which is only introduced in the next paragraph. Particularly the sentence "Indeed, the Arg416Ser variant abolishing GSH inhibition results in a constitutionally active transporter" only makes sense in light of the hypothesis that abolishing GSH binding through this mutation subsequently prevents association of the RCK domains and inhibition of the transporter. It would be better understandable if this section were moved to the next paragraph, after this interaction hypothesis has been introduced.

Thank you, we have moved this sentence to the next section

-

- Line 304: 'KdpFABC'

Changed

- Line 335: change 'conformations' to 'conformational changes'; change 'substrate bound ion' to 'substrate binding site' or 'ion binding site'

Sentence has been modified

- Line 369: I would be careful describing the alphafold prediction as a conformational state – particularly for multi-domain proteins, AF has a tendency to predict the individual domains with high accuracy, while moderately flexible to flexible linkers are often predicted in a random orientation with a high uncertainty score, as is the case for the KefC prediction.

We have now clarified that KefF is required for KefC activity. The most plausible explanation is that KefF forms a complex with KefC and this will require a likely detachment of the RCK domain, which are currently restricting movement of the

transport domain in the KefC Cryo-EM structure. The AlphaFold predicted model is consistent with a “state” where the RCK domains are exposed to interact with KefF. The linker domains have poor confidence as they are likely flexible, which is consistent with our inability to model them in the Cryo-EM maps.

- Line 990: Capitalize ‘C’ in ‘KefC’

Changed

- In general: Check hyphenation, things like ion transporters (line 20), ion selectivity (line 38), distantly related (line 51) for example need not be hyphenated

Changed

Reviewer #2 (Remarks to the Author):

The manuscript by Gulati et al. addresses Kef transporters belonging to the CPA2 superfamily. Kef transporters are K^+/H^+ antiporters and are part of the diverse biological response bacteria have to maintain potassium homeostasis.

The manuscript presents two cryo-EM structures of the bacterial K^+/H^+ antiporter KefC. In addition, the purified KefC protein in complex with AMP has been reconstituted into proteoliposomes and used for Solid Supported Membrane (SSM) based electrophysiology. Finally, the manuscript presents some MD to support the identification of a spherical density in the expected binding site as potassium.

Several findings in the manuscript are of interest to the community. This represents the first structure of a Kef K/H transporter. KefC has a C-terminal RCK domain and the manuscript presents the first glimpse of how this regulatory domain interacts with the transporter domain. In addition, the structures reveal a spherical density that is very likely a potassium ion, and thus represent the first substrate bound structure from the CPA2 superfamily.

The observation that a central crossover in the transmembrane part is shorter than in other CPA2 members and how this related to substrate recognition and specificity is very interesting.

For me, the key point of the manuscript is the regulation/inhibition of a CPA2 transporter by a RCK domain, very similar to the regulatory domains found on potassium channels such as TrkH. This is best exemplified by the AMP/GSH complex structure.

The structural data is convincing. The SSM data less so in its present form. I am not an MD expert and will refrain from commenting on that except to say that it would be beneficial to use a bit of time to better describe the aim and methodology of this part for the non-expert (e.g. at line 181). The methods are well described in general otherwise.

Major points:

Is the AMP-only sample an inhibited state or an active state? This is not discussed anywhere and is a major omission.

This is an essential point and we have clarified this in the re-submission. As explained in our responses to referee 1, the RCK domains require AMP for the dimerization of KefC. We demonstrate that if we remove all AMP in our purified sample of KefC in detergent by incubating with shrimp alkaline phosphatase for 30 mins on ice, then the protein falls apart into monomers. It is surprising to us that the KefC dimer is so unstable when the RCK domains dissociate and confirms that in the presence of AMP, these RCK domains are forming stable interactions to KefC. Unfortunately, we have been unable to reconstitute an active system in proteoliposomes to measure actual ion-exchange (turnover).

As explained to referee 1 (Q5) full activation of KefC requires the soluble protein KefF. These conclusions were drawn from cell-based assays using specially-constructed *E. coli* strains and atomic absorption spectroscopy. We have purified KefF, but we are still unable to activate KefC transport *in vitro*. It is possible that we still need to add GSX to achieve activation.

To conclude, it seems clear that RCK detachment is required to remove auto-inhibition of KefC, but it is unclear if activation is by i) AMP displacement or ii) and/or binding of KefF to further stabilize the detached RCK domains, in combination of binding of GSX. We now think the later model is the most likely (however unconventional) since the mutation of AMP coordinating residues abolishes KefC activity in cells (Biochemistry. 2017 Aug 15; 56(32): 4219–4234 [REDACTED]).

Briefly at least four states are implied by the study as I understand it:

1) AMP-only bound: The first structure presented and the sample used for SSM. Is this sample active or inactive? A comparison (superposition, RMSD) to the AMP/GSH structure is completely lacking and this should be rectified. Presumably this AMP-only form is not physiologically relevant (based on the intro and fig 1a)?

That's a good point. Based on referee 1 comments we have realized we have not properly introduced the extensive work on the structures and biochemical characterization of RCK domains, which have been done previously. We show that the structures with either AMP or AMP/GSH are essentially the same (r.m.s.d. 0.5 Å), but the GSH fills a positively charged interface. Given the high concentration of GSH present in the cytoplasm then, in fact, the structure of KefC with AMP and GSH is physiologically the most relevant. Since we have both structures, however, we have included them here and clarified this point in the text; also, nice to confirm that GSH addition doesn't cause an obvious conformational change.

Superimposed complexes of KefC, with AMP (cyan), AMP and GSH (orange) and D156N with AMP (grey) (Supplementary Fig. 10c).

As replied to referee 1, the SSM data is most consistent with ion binding rather than transport and updated the manuscript.

2) AMP and GSH bound (biologically relevant): This is actually the most interesting structure in the manuscript but is only mentioned at the end. This should be the main finding of the paper in my opinion, as we now have a biologically relevant and well-defined inhibited structure.

That is a fair point. Because the structures of AMP and AMP/GSH are so similar we think current paper order still works well. Previously, it has been shown that GSH does not show any further stabilization of the soluble RCK domains in addition to AMP (Biochemistry. 2017 Aug 15; 56(32): 4219–4234). Based on 3DVA of the cryo-EM

structure for KefC with AMP we see that the RCK domains are quite dynamic and its possible that the final reconstitution represents the most energetically stable conformation, which is the inhibited one. Rather than GSH driving a new conformational change we think it helps to stabilize the predominant state that we are modelled with AMP present only. The 3DVA of KefC in complex with AMP/GSH shows that the GSH density is clearest when the RCK-domain and transporter module interactions are strongest. Indeed, we now highlight that an arginine residue bridges an interaction between the bound GSH to $\alpha 7$, the main helix interacting with the transporter module. Consistently, K^+/H^+ exchanger homologues that are not thought to be regulated by GSH do not have the $\alpha 7$ helix (see figure to Referee 1, Q1) (*Membranes* **2023**, 13(5), 465).

3) GS-X only bound: This is presented as the physiologically active form. Note that fig 1a (incorrectly?) show that GS-X and AMP bind at the same time. This is at odds with e.g. line 367.

It seems, in fact, that GS-X may not compete out AMP binding (*Biochemistry*. 2017 Aug 15; 56(32): 4219–4234) and they both “may” be able to bind at the same time, as concluded from thermal-shift analysis of purified KefC RCK domains. In light of these experiments that we had missed previously, we have updated Fig. 1.

DSF experiment to determine the effect of AMP and GSX on SdKefQCTD
(Biochemistry. 2017 Aug 15; 56(32): 4219–4234 [REDACTED])

4) apo-form, nothing bound: This is made in the study by dialyzing away the AMP. This sample is not physiologically relevant but is expected(?) to mimic the GS-X active form, based on the observed mobility of the RCK domains.

We were unsure. Our stability studies indicate that AMP removal will result in RCK detachment. The poor 3D reconstruction of the KefC sample after AMP dialysis could either be because i) we have only a small fraction of KefC stable protein that still retains AMP or ii) that the RCK domains have detached and, as such, the KefC transporter is more dynamic. In line with the second reasoning, comparing 2D classes with a similar number of particles with AMP, shows that we cannot clearly see features for the RCK domain when AMP was dialysed away during sample preparation in the 2D classes, i.e., arguing against that is just a small fraction of AMP-retaining KefC.

Nevertheless, while supporting an autoinhibition model based on RCK interactions to the transporter, since full activation of KefC requires the KefF protein (see responses to referee one), we now think that it does not represent an active state of the KefC protein.

2D class comparison of KefC protein with AMP added (left) and after dialysis to remove AMP (right)

The SSM is done using the AMP-only sample. Assuming that this is similar to the AMP/GSX structure it should be inactive. Yet SSM shows a signal. Do the authors believe that the signal is from the half-reaction (as implied, e.g. line 199-201) or from

the binding of K⁺ to the inhibited protein? This need to be clearly addressed and explained and related to the expected mode of inhibition (presumably inhibition is by locking the transporter in the inward state?).

Excellent point. We have now modified the text to clarify that the SSM-data is more consistent with binding rather than a translocated charge as we had initially proposed, i.e., based on the LPR ratio data and the lack of increasing time decay (see response to referee one). This also makes more sense with the updated stability analysis with AMP and the relatively weak peak currents observed. Rather than the removal of AMP, we now think that we need to form a complex with KefF to fully activate KefC. Unfortunately, we have been unable to generate this complex in vitro.

J Bacteriol. 2000 Nov; 182(22): 6536–6540.[REDACTED]

Is the domain swap of the RCK domains a relevant physiological feature or a sample artifact? This should be discussed in the manuscript.

We have clarified in our responses its well known the RCK domains are typically domain-swapped via $\alpha 6$ of the Rossmann fold (Structure 17:893-903). Presumably it is domain-swapped to enable the RCK domains to detach together *en bloc*. Interestingly, the additional helices ($\alpha 7$) are making most of the inhibitory contacts. Furthermore, the KefF protein does also bind to the dimeric RCK domains. (PNAS 114: E1101-E1110).

The fact that the two RCK domains are not binding the transporter monomers in a symmetrical fashion is very interesting but is not discussed. Why is binding asymmetrical? More effort should be made to explain how inhibition is achieved in general. Do you have any clues from the structure. Do you think the monomers can transport K⁺ individually or is transport in the dimer coupled. Do you think that both interactions from the RCK domains to the transporter domains lead to inhibition, or is only one monomer inhibited in the presented state?

As replied Referee 1, although the RCK domains interact with the transporter module asymmetrically, it appears to be a fairly stable conformation as observed by the same structural state for three different preparations of KefC (WT+AMP, WT+AMP+GSH and D156N), as well as the similar local resolution estimates of the RCK domains as the

transporter domain. Is this RCK asymmetrical binding to the transporter module functionally important? We think so, as we now show that single point mutations in the transporter-RCK domain interface are able to dramatically influence the stability of KefC, i.e., mutations can either retain more or less KefC homodimer after heating than WT. It is possible that the reason for this is that the RCK domain needs to detach to remove inhibition, which we believe enables the KefF protein to interact with these domains. In fact, we do have an additional 3D reconstruction where the RCK domains are positioned symmetrically relative to the transporter module. In this case, $\alpha 7$ is rotated away from the transporter module in both RCK domains. Surprisingly, the map density for the transporter module is very poor, especially for the linker helix TM7 and the dimeric interface. So, it seems that symmetric RCK domain interactions might lead to a less stable state. Indeed, even in the high-resolution reconstruction, the linker helix TM7 is more ordered in the protomer where it is interacting with $\alpha 7$ in the RCK domain as compared to when it is not. We have now mentioned this reconstruction in the manuscript, but we do not discuss these maps in detail as its unclear if this poor reconstruction is physiologically/relevant or not. Lastly, given the RCK domains are domain-swapped we think both transport domains are inhibited in the current state, which would be further consistent with the SSM data analysis that is now presented.

The statement Line 203-206 seems confused or perhaps even wrong? Electroneutral

transport cannot be distinguished from electrogenic transport by the directionality of the peak. The directionality shows if a positive change is going into or out of the proteoliposome. E.g if transporting a negative substrate, the current would be negative.

Apologies, we meant that the peak direction can distinguish between electroneutral and electrogenic "Na⁺/H⁺" exchangers. Given we now interpret the SSM-data as binding we have removed this statement.

L201: Ph 7.8 is mentioned vs fig 2d legend that mention pH 8.5?

Thanks for spotting this error. We have corrected to pH 7.8 for all measurements.

L204: "in the presence of 200 mM NaCl" vs the figure legend where 30mM KCl is mentioned? There are many similar omissions and inconsistencies in the text and figure legends.

OK, these have now been corrected.

I agree that the current peak observed likely represents binding and not transport. However, this is not demonstrated, and could readily be done by measuring the full peak width at half maximum (FWHM) using different Lipid to Protein Ratios (LPR). If we only observe binding, then FWHM should be independent of LPR. If transport is involved, the FWHM will increase with increasing LPR. For an example of this type of analysis see for instance figure 1 in Bazzone, A., Zabadne, A.J., Salisowski, A., Madej, M.G., and Fendler, K. (2017). A loose relationship: Incomplete H⁺/sugar coupling in the MFS sugar transporter GlcP. *Biophysical Journal* 113, 2736-2749.

Yes, we agree and these experiments have now been included.

Some suggestions. Have you tried this:

Try the apo-form for SSM (i.e.. dialyze away the AMP). What happens?

While making proteliposomes our sample is diluted without AMP and we have a PD10 step to remove cholate and probably AMP in the last step. However, since our SSM-data is more consistent with binding rather than transport, it is unclear if we are only detecting currents for protein that retains AMP. We have added AMP during liposome preparation and the peak currents and K⁺ binding affinity are similar, which would be consistent with this conclusion.

If you think the SSM sample is inactive, add GS-X to activate. What happens?
 We haven't been able to get hold of GS-X and we are currently attempting to synthesize adduct in house to investigate this.

If you believe the SSM sample is active, add GSH to inhibit. What happens?
 We no longer believe the KefC protein is active under these conditions.

L282: Arg416Ser is constitutively active. This mutant should be tested with SSM.
 What happens?

The purified R416S variant shows similar peak currents as the structural KefC construct. This indirectly supports that KefF (and likely GS-X to form the complex) is needed and what we measured by SSM is only binding.

Could you make a truncated KefC without the RCK domain and test with SSM?

We have tried, but the protein cannot be purified due to instability (see supplementary Fig. 5c).

In Fig2d the D156N and 'empty' traces are similar and negative. In sfig6 the empty trace is missing, but the D156N peak has reversed its direction and is now positive and similar to the wt. Explain this observation. Also, the empty trace must be added here. Are you sure the D156N mutant doesn't also bind K^+ to some extent?

We initially added 200 mM NaCl in the inactivating buffer and 30 mM KCl in the activating buffer. However, the high-salt is making the liposomes leaky and we probably see non-specific ion-flux (negative current) even in empty liposomes. We find that the liposome integrity is better maintained with Choline Chloride in the inactivating buffer. For sake of clarity, we have no longer included these optimization traces.

Comparing sfig6a and sfig6c: sfig6c is used to demonstrate that the mutant and wt are indifferent to other ions. without including K^+ in the same analysis (on the same sensor) this is meaningless. In all cases of sfig6c the current at 25 mM Ion is around 500 pA. This is perfectly comparable to the peak current of K^+ shown in sfig6a. At present it seems that the KefC is binding or transporting all of these ions regardless of the D156N mutation or not. Showing (from the same sensor) that K^+ is the only case where a clear difference can be seen between mutant and wt, and also that K^+ elicits a higher current response than the other ions is a necessary control.

If we focus just on K^+ we see that there is a clear difference for the peak responses and the affinities measured for WT vs ion-binding variants. We are able to confirm K^+ binding to KefC as further supported by LPR data.

Na^+ and Li^+ show similar strength peak currents as K^+ addition to KefC, but there is no difference between these signals and the dead D156N variant; this aspartate is strictly conserved cross all Na^+/H^+ exchangers and is well known to abolish ion binding (Nature Comm 9:4205 (2018)). Moreover, if we extended the concentration range to 100 mM then it becomes clearer that the K^+ responses remain saturated, whereas the LiCl and NaCl keep increasing. The measured K_d for K^+ is (10 mM), Na^+ (60 mM) and Li^+ (457 mM), which could be interpreted as KefC can bind Na^+ and Li^+ weakly, but with subtracted back-ground then there is no clear binding for either NaCl or LiCl.

In the beginning we compared KCl vs LiCl responses on the “same” sensors rather than different sensors. We didn’t see any differences and given the small variation between independent sensors, we concluded that it was a more robust approach to use multiple independent sensors. Please see an example below with titrations on the same sensors.

Minor Points:

The manuscript was somewhat difficult to read and has a range of unclear and confusing statements that should be addressed. Some examples:

L93: active pH. What does this mean?

Because all Na⁺/H⁺ have a conserved aspartic acid in their binding site as well as other charged residues, Na⁺ binding is pH dependent. Given H⁺ ions are also transported, there is a bell-shaped ion-exchange activity based on simple competition between Na⁺ vs H⁺. In the Na⁺/H⁺ antiporter field the term active pH, refers to the pH where the protein can catalyze ion-exchange, in contrast to inactive pH, where no transport is possible as the ion-binding site is protonated.

In order to avoid confusion, we have removed the term “active”.

L94: “images” This is micrographs or movies.

Changed

L138: K307 is on TM10, but in fig 2b it is on TM11?

Thanks for spotting. When we first published NapA we labelled the first helix TM -1 as NhaA has one less helix and everyone in the field was familiar with the NhaA transporter fold as “the” model system. It turns out that NhaA is more of an outlier and the majority of Na⁺/H⁺ exchangers have 13 TMs and we now just refer to the first helix

as TM1, and not TM -1. So we should have written that K307 is on TM11 and not TM10.

L139-145. The intermittent discussion of specific NapA residues here and elsewhere makes the text hard to read. Perhaps move to a discussion later?

Thank you for the suggestion. We have moved some of this text.

For both cryo-EM datasets, very large datasets were collected (up to 25K movies and 7-10M particles picked), but in the end only ~300K particles were used for the reconstruction. It would be interesting to discuss this a bit. Is it possible that other conformations of KefC (e.g. symmetrical conformations) were present in the data for instance?

As now clarified in response to an earlier question we do have an additional conformational with symmetrical RCK-domain-transporter interactions, but due to poor map quality for this reconstruction we have not included the model.

Title: Definition vary in the literature, so feel free to ignore, but the word exchanges should be reserved for transporters that exchange compounds (e.g ADP/ATP exchanger or the CMP/CMP-sialic acid exchanger), while secondary active transporters that are driven by sodium or proton gradients are antiporters (or symporters depending). In the manuscript the word exchanger is only used in the title, not the main text where antiporter or transporter is used.

It's a good point. I think this was probably the reason why bacterial Na⁺/H⁺ transporters have been referred to as "antiporters", whilst the mammalian ones have been referred to as "exchangers". That is, the bacterial ones are mostly electrogenic and so H⁺ gradients drive Na⁺ uphill, whilst the mammalian ones are electroneutral and so the ions are going down-hill in both directions and are therefore "technically" ion-exchangers.

Perhaps exchangers is the safer terminology and probably more accurate since KefC is electroneutral, but I am also fine with using the antiporter terminology if you think this would be required.

Abstract: Only one structure is mentioned but two are presented.

The abstract has been updated to mention all included cryo-EM structures.

Intro: be clear that K⁺ transport is driven by proton transport. Discuss references on stoichiometry. What is known about this for Kef systems?

Based on the structure and previous in vivo functional data, KefC and homologues are electroneutral K⁺/H⁺ antiporters. In similar Na⁺/H⁺ antiporters it is a 1:1 exchange.

So it depends. With a S_{in}/S_{out} ratio of 200 mM/20mM K^+ then there is a 10-fold concentration difference. In contrast a pH difference of about 1.5 (pH 7 vs pH 5.5) is ~30-fold. So, presumably in many conditions the pH gradient drives K^+ efflux, but if the pH differences is only 1, then transport could be driven by the K^+ gradient instead.

Summarize the known structures of RCK domains. Do we have structures with AMP and/or GSH and GS-X bound. This is unclear, but seems very relevant.

Yes, we have these structures and they have now been included.

L53, 55, 56, 75 and elsewhere. When comparing to other protein families and proteins it would be instructive to have the sequence identify mentioned somewhere. At least for a few cases.

This information has now been added to the sequence alignments.

L114: The lipid identity is not super relevant for the findings, and the whole section could be shortened. The identity is speculative based on thermostability only. IS POPG found in E coli membranes, and is it abundant?

We respectively disagree. We have found specific lipids to be very important for oligomerization and function in a number of related Na^+/H^+ exchangers. We have shown that ligand-like lipid binding can be assessed by thermal-shift assays that we have developed, including cardiolipin binding to the Na^+/H^+ antiporter NhaA (Nature Comm. 9:4253 (2018). E. coli lipids are roughly 75% PE, 20% PG and 5% CL (J.Bact. 199:e00849).

We initially determined the cryo-EM structure of KefC in nanodiscs, but made with POPC lipids and found that the dimer interface had collapsed. We know that without the right lipid that KefC is unstable, but it is unclear if the PG lipid has a regulatory role in addition the just a structural role.

L123 An RMSD(Ca) of 4.3 Å seems very high. This number needs to be better explained. In fig 1e it doesn't look like a 4.3 Å RMSD deviation. Maybe our methods to calculate RMSD differ?

We used the “align” command of pymol to superpose these two structures that resulted in the alignment with RMSD of 4.3 Å, but using the “super” command the reported RMSD is a bit lower at 3.6 Å (visually the look the same). We have updated the manuscript to report 3.6 Å and we better highlighted the regions that are poorly aligned.

L150: you could discuss if the coordinating residues are conserved? I assume so. Does the observed coordination fit with potassium. Related in fig 2c it is impossible to judge the density. What is the local resolution?

Thank you for raising this valid concern. As shown to Referee 1 we have included additional KefC structures to support the assignment of the K⁺ ion. Shown below is the consurf analysis performed on kefC and as expected all the residues important for the binding of K⁺ ion are highly conserved. We have also included a supplementary figure that includes more surrounding density around the potassium binding site to help judge the quality of the data.

ConSurf analysis of *E.coli* KefC using default parameters of the ConSurf server. **a.** Sequence conservation annotation for the transporter module. Residues important for K⁺ binding are boxed. **b.** Sequence conservation annotation for RCK domain. Residues important for AMP and GSH binding are marked with stars and triangles, respectively.

Map density covering the 12 Å radii surrounding area around the potassium binding site. Maps were rendered in pymol with a sigma threshold of 7.2

L229, DALI hit and “most similar”: Everything is most similar to something else. What is the Z-score? What is the RMSD?

We have answered this question in detail in response to the first referee’s question number 1.

L237: compare the structure of the RCK domain to solved structures. It seems like a strange omission to not do this?

Thanks for spotting this oversight. We have updated our paper where we have compared the RCK domains to solved structures

L352-360: NHE regulation does not seem that relevant and could be shortened. Maybe it would be more relevant to compare to the TrkH regulation from similar regulatory domains. Do these domains work in a similar fashion?

We now clarify that connection between the RCK domains and the linker helix TM7. In related NHE proteins we see a similar interaction between the soluble regulatory domains and this helix. We hope this discussion makes more sense now. We think the TrkH regulation is different to reflect the differences between transporters and channels.

L384: Since you have KefF purified, try to add it to SSM to see if this improves/changes results?

Very good point. Since we are still trying to obtain a functional KefC-KefF complex, we will perform this experiment in the follow up study, but our initial experiments showed no clear differences.

Table 1: cryo-EM GSH model: Out of curiosity, why did you do manual map sharpening for the AMP/GSH dataset?

The output from Phenix Autosharpen was very similar to map sharpening done in Coot and we didn't reflect too much on this.

fig 1a: This panel could be improved. e.g. arrows showing K and H transport are not clear. Is AMP always present, also with GS-X? From the text it seems like this is not the case.

We have now updated Fig. 1a as referee 1 had the same concerns.

fig 1e: Labels are needed to orient the reader.
labelled

fig 2b: label TM6 also.
Labelled

fig 2c: It is very hard to see what is happening here with the density. Also label key helices.

Thank you for your suggestion. Key helices have been labelled in the figure

fig 2e: no units on Kd value.
Added

fig 3d: is very hard to read. Try to link it to panel c perhaps?
We have modified the figure extensively

fig 2a & 3e: you should to define the blue-to red range of your electrostatic potential calculations (in $kT\epsilon^{-1}$).

fig 4: Again, somewhat confusing. E.g. only one AMP is shown but each monomer binds AMP. Also, now AMP is gone from the GS-X activated state (in contrast to fig 1a).

Fig. 4 has been re-made again, which we hope is now clearer.

Sfig2: Only one curve is shown for the FSC. Which one? You should preferably present "no mask", "spherical", Loose "tight" and "corrected". For a discussion see e.g. Chen, S. et al. High-resolution noise substitution to measure overfitting and validate resolution in 3D structure determination by single particle electron cryomicroscopy. Ultramicroscopy 135, 24–35 (2013).

Thank you so much for pointing this out. We have previously shown only the corrected FSC curve. We have now replaced the figure with all the necessary traces

Reviewer #3 (Remarks to the Author):

The authors report on the structure determination of the K^+/H^+ transporter KefC through cryo-electron microscopy. The mechanism of transport is also inferred by combining structural data and other experimental techniques, including electrophysiology. The study is complemented with Molecular Dynamics simulations aimed at elucidating the determinants of selectivity for K^+ versus Na^+ . The paper is very clearly written, and in my opinion, is an excellent piece of work overall. Unfortunately, concerning my field of expertise, I have some concerns regarding how the free energy calculations have been performed. Specifically, I am rather confused about whether the authors performed the $K^+ \rightarrow Na^+$ transformation using equilibrium or out-of-equilibrium simulations. I am pretty sure that the plots reported in Figure S5 are work profiles, but the authors employ terminology that is consistent with equilibrium simulations, like “FEP” and “windows”. In particular, term windows should be reserved for staged calculations along the perturbation. Instead, to my best understanding, the authors extracted 100 configurations and for each of which they performed an out-of-equilibrium switching, am I right?

We thank the reviewer for suggesting the change in the text to use consistent terminology to help the reader orient themselves. We eliminated the word ‘windows’ since the calculations are indeed out-of-equilibrium simulations. We did, indeed, run one MD simulation (and during the review process 2 additional ones) and extracted 100 snapshots on which we performed the out-of-equilibrium switching. We amended the methods section to clarify this point.

Also, it is not clear which estimator the authors employed, as even though they refer to the Bennet Acceptance Ratio, the plots shown in Figure S5 rather suggest that the work distributions were used according to Crooks’ theorem.

The plots are indeed from the Crooks Fluctuation Theorem, we calculated the $\Delta\Delta G$ both with the Bennet Acceptance Ratio and the Crooks Fluctuations, we now report just the Crooks Fluctuation in the supplementary figure in order to maintain consistency between figures and text.

I would strongly advise the authors to provide some clarification of their computational setup. Please notice that this is not (only) a matter of being picky regarding the proper terminology, but it is also related to the reliability of results. Indeed, if the free energy difference was estimated using out-of-equilibrium techniques, 50 ps of simulation time might be enough to obtain meaningful results (with a huge number of realizations), but in my experience, these simulations are way too short in the case of equilibrium runs. In any case, authors are encouraged to increase the length of their simulations to provide an indication that the estimated free energy difference is close to convergence.

As pointed out in the first response and now more clearly delineated in the methods section, the simulations comprise an out of equilibrium switching between potassium and sodium (and vice versa). Due to this, the convergence of the simulations is not

directly addressable by just elongating the simulations but we need an estimation of how reliable these results are too.

Accordingly, we followed the reviewer's advice and increased the time of switching to 100, 200 and 500 ps and these results are summarized in the new figure; moreover, we added two additional replicas and performed the switch separately for each of the ions in the two protomers, resulting in six independent replicas for each calculation. The results are consistent with our previous estimation.

Apart from that, the free energy difference for the $K^+ \rightarrow Na^+$ transformation in water should be closely related to the solvation-free energy difference between the two ions. Are results consistent with previous estimates?

The solvation free energy difference between the two ions can indeed be calculated similarly, adding though the switch between the two in void. The results we obtained are compatible with previous estimate of K^+/Na^+ selectivity in channels (<https://doi.org/10.1016/j.bpc.2006.05.033>) and transporters ([10.1021/acs.jpcc.9b08552](https://doi.org/10.1021/acs.jpcc.9b08552)).

Maybe it's me, but how does the reported free energy difference between the two ions of 15.17 kJ/mol compares with the values reported in Figure S5? (-139.90 and -170.23)

The reviewer is correct in pointing out that the values were not properly explained, we added a section in the supplementary figure to show where the numbers come from. In brief, we initially performed the transformation of both of the ions at the same time, calculated the corresponding $\Delta\Delta G$ and then divided the result by two. The reviewer comment showed us that this approach was not properly explained in the text and we realized that a more accurate estimate could be achieved by treating each ion individually, so now we switch each individual ion independently and report the mean of the 6 resulting values (2 ions X 3 replicas).

Please provide more details regarding the simulation setup, like the water model, cutoffs employed, long-ranged electrostatics, timestep, and so on. This is very important information for ensuring the reproducibility of results.

We prepared the systems with CHARMM-GUI and did not modify the parameters chosen for the equilibration. However, it is indeed better to report directly the chosen parameters without relying on the accessibility of an online tool. We thus added the water model and the time step used, thermostat and barostat information in the methods section.

Supplementary Fig. 8a-b - **A** Calculated ΔG for the K^+ to Na^+ transition in protomer A (circles), protomer B (crosses) and water (triangles) for a switching time of 50ps, 100 ps, 200 ps and 500 ps replicas are shown in lighter colors, mean in darker colors and as squares ; **B** Thermodynamic cycle reporting the calculation for the $\Delta\Delta G$ of K^+ to Na^+ transition; **C** Calculated $\Delta\Delta G$ for the K^+ to Na^+ transition, as obtained by the difference between the ΔG of the reansion in water and in the protein.

REVIEWERS' COMMENTS

Reviewer #1 (Remarks to the Author):

I would like to thank the authors for their careful revision of the manuscript and also for sharing findings from experiments that did not work and that one obviously does not include in a publication. These were extremely helpful to me in understanding some of the limitations of functional analysis. In my opinion, the manuscript has improved massively and I am now happy to recommend it for publication in its present form. Congratulations on this beautiful work from my side!

Reviewer #2 (Remarks to the Author):

The authors have addressed all my concerns.

The paper is significantly improved, the new dataset with D156N showing the lost of potassium density is clean. The alkaline phosphatase experiment is also quite elegant showing that AMP is required for stabilization of the dimer. More control SSM experiment has been added, and the new analysis regarding RCK flipping in the AF2 model is interesting.

Reviewer #3 (Remarks to the Author):

Unfortunately, it seems the authors have only partially addressed my concerns.

Despite claiming in the rebuttal letter to have introduced information regarding timestep, thermostat, barostat, and water model; I cannot find any evidence of that in the manuscript. There is also no mention of the composition of the membrane used in the simulation. Even the information regarding the force field used is missing!

Furthermore, while the manuscript refers to FEP calculations, the rebuttal letter clearly mentions out-of-equilibrium simulations. Similar inconsistencies are found regarding the word "window"... Could it be that the computational part of the Methods section was inadvertently omitted from the updates?

On the other, hand the authors convinced me regarding the robustness of their simulations.

R

Unfortunately, it seems the authors have only partially addressed my concerns.

Despite claiming in the rebuttal letter to have introduced information regarding timestep, thermostat, barostat, and water model; I cannot find any evidence of that in the manuscript. There is also no mention of the composition of the membrane used in the simulation. Even the information regarding the force field used is missing!

Furthermore, while the manuscript refers to FEP calculations, the rebuttal letter clearly mentions out-of-equilibrium simulations. Similar inconsistencies are found regarding the word “window”... Could it be that the computational part of the Methods section was inadvertently omitted from the updates?

On the other, hand the authors convinced me regarding the robustness of their simulations.

Please accept our sincere apologies for the confusion. Yes, we accidentally submitted the manuscript version in which the method section for simulation work had not been updated. This has now been rectified.